# Lysine/RNA-interactions drive and regulate biomolecular condensation

Tina Ukmar-Godec[1,2], Saskia Hutten[3], Matthew P. Grieshop [4], Nasrollah Rezaei-Ghaleh[1], Maria-Sol Cima-Omori[2], Jacek Biernat[5], Eckhard Mandelkow[5,6], Johannes Söding [4], Dorothee Dormann [3,7] & Markus Zweckstetter[2,8]

Cells form and use biomolecular condensates to execute biochemical reactions. The molecular properties of non-membrane-bound condensates are directly connected to the amino acid content of disordered protein regions. Lysine plays an important role in cellular function, but little is known about its role in biomolecular condensation. Here we show that protein disorder is abundant in protein/RNA granules and lysine is enriched in disordered regions of proteins in P-bodies compared to the entire human disordered proteome. Lysine-rich polypeptides phase separate into lysine/RNA-coacervates that are more dynamic and differ at the molecular level from arginine/RNA-coacervates. Consistent with the ability of lysine to drive phase separation, lysine-rich variants of the Alzheimer's disease-linked protein tau undergo coacervation with RNA in vitro and bind to stress granules in cells. Acetylation of lysine reverses liquid–liquid phase separation and reduces colocalization of tau with stress granules. Our study establishes lysine as an important regulator of cellular condensation.

[1] Department of Neurology, University Medical Center Göttingen, University of Göttingen, Waldweg 33, 37073 Göttingen, Germany. [2] German Center for Neurodegenerative Diseases (DZNE), Von-Siebold-Strasse 3a, 37075 Göttingen, Germany. [3] BioMedical Center (BMC), Ludwig-Maximilians−University Munich, 82152 Planegg-Martinsried, Germany. [4] Quantitative and Computational Biology Group, Max Planck Institute for Biophysical Chemistry, Am Faßberg 11, 37077 Göttingen, Germany. [5] German Center for Neurodegenerative Diseases (DZNE), Venusberg-Campus 1, Gebäude 99, 53127 Bonn, Germany. [6] CAESAR Research Center, Ludwig-Erhard-Allee 2, 53175 Bonn, Germany. [7] Munich Cluster for Systems Neurology (SyNergy), Feodor-Lynen-Straße 17, 81377 Munich, Germany. [8] Department for NMR-based Structural Biology, Max Planck Institute for Biophysical Chemistry, Am Faßberg 11, 37077 Göttingen, Germany. Correspondence and requests for materials should be addressed to M.Z. (email: Markus.Zweckstetter@dzne.de)

Non-membrane-bound compartments, which enable the coordination of intracellular reactions and processes, form and dissolve in response to cellular signals[1]. The formation of non-membrane-bound compartments occurs predominantly through liquid–liquid phase separation (LLPS) of macromolecules[2–4]. As a result of their liquid, gel-like, or solid nature these phase-separated compartments are commonly referred to as biomolecular condensates[1,5]. Intracellular phase separation is driven by weak multivalent interactions between proteins, typically containing disordered, low-complexity regions, as well as RNA and DNA[5–11]. Important forces driving biomolecular condensation include cation-π interactions between tyrosine from prion-like domains and arginine from RNA-binding domains[7], as well as coulombic interactions with negatively charged phosphate groups of nucleic acids[10,12,13]. The molecular properties of phase-separated compartments are further modulated by glycine, which enhances droplet fluidity, and glutamine and serine, which promote droplet hardening[7]. In agreement with an important role of post translational modifications in the regulation of biomolecular condensates[14], LLPS is modulated by phosphorylation of tyrosine[15] and serine[12,16], and methylation of arginine[17–20]. One of the most highly post translationally modified amino acid is lysine[21]. Despite the importance of lysine for cellular function[21], little is known however about the role of lysine in biomolecular condensation.

Lysine with its lysyl (($CH_2$)$_4$$NH_2$) side chain is a basic and positively charged amino acid at physiological pH. It influences the function of proteins involved in development, cell–cell interaction, signal transduction, and many other biological processes[21]. In addition, lysine is often involved in histone modifications and thus gene expression[22]. The ε-amino group of lysine participates in hydrogen bonding and acts as a general base in catalysis. The lysine side chain can be reversibly modified by acetylation and other covalent modifications[21]. These modifications generate binding motifs[15], or increase/decrease the net charge of the protein and thus tune the strength of biomolecular interactions[17]. Post translational modifications of lysine side chains are catalyzed by hundreds of enzymes and may lead to changes in the activity and intracellular localization of proteins[23].

The microtubule-associated protein tau is expressed in neurons and aggregates into neurofibrillary tangles in the brains of patients with Alzheimer's disease[24]. Tau is an intrinsically disordered protein[25,26] and post translational modifications of tau modulate tau/microtubule-interaction, tau localization, deposition, and neurotoxicity[27]. Alternative splicing produces six different isoforms of tau in the human central nervous system[28]. The six tau isoforms differ in the number of N-terminal inserts and have either three or four imperfect repeats in their C-terminal half[29]. Lysine is one of the most abundant amino acids in tau, can be acetylated by the acetyltransferases p300 and CBP, and deacetylated by sirtuin 1[30]. Tau also has intrinsic acetyltransferase activity[31]. Acetylation of tau is elevated in conditions of cellular stress[30], dysregulates tau homeostasis due to acetylation-mediated blockage of tau polyubiquitylation[30], interferes with binding to microtubules[32] and promotes synaptic dysfunction[33].

Tau can undergo LLPS and form liquid-like droplets[34–37]. The strong increase in tau concentration within phase-separated tau droplets enhances tubulin polymerization, recruits negatively charged factors and promotes misfolding and aggregation[34–37]. Binding to RNA and hyperphosphorylation promote tau phase separation[34–36]. Tau pathology co-localizes with stress granules (SG)[38], cellular condensates that have been linked to neurodegenerative disease[39]. Tau accelerates SG formation, RNA-binding proteins co-localize with tau inclusions[40], and interference with SG formation modulates tau pathophysiology[41].

Here, we study the contribution of lysine to protein phase separation and biomolecular condensation. Bioinformatic analysis shows that lysine is enriched in disordered regions of proteins in cytosolic protein/RNA granules when compared to the entire human disordered proteome. We then use synthetic peptides with distinct amino acid content to study molecular properties of phase-separated states formed by lysine- and arginine-rich peptides and show that different lysine-rich sequences of the protein tau undergo complex coacervation with RNA and bind to SGs. We find that acetylation reverses lysine-driven peptide and protein LLPS and decreases association of tau with SGs. Our work identifies lysine as an essential molecular determinant of biomolecular condensation.

## Results

**Lysine is enriched in disordered regions of cytosolic granules.** Processing bodies (P-bodies) and SGs are non-membrane-bound protein/RNA granules in the cytosol[14]. They contain at least 100 proteins, which interact with RNA[42,43]. To gain insight into the nature of weak multivalent interactions in protein/RNA granules, we predicted disorder with IUpred[44] for three protein sets: the entire human proteome, the proteins associated with P-bodies and those associated with SGs, as measured by proteomics[42]. We found that disordered regions are highly abundant in SGs and P-bodies. Whereas 25% of amino acids in the total human proteome are predicted to be disordered, the percentage is 35% for the SG proteome and 38% for P-bodies. To see the relative importance of disordered regions and their distribution across sequences, the percent of residues predicted to be disordered in each protein in the respective proteomes was plotted (Fig. 1a, Supplementary Fig. 1a and Supplementary Data 1–3). Compared to the total human disordered proteome, disordered regions in P-body-associated proteins are most strongly enriched in K, a factor 1.3 above the human disordered proteome ($p$-value = 1.5E-28, one sided t-test). Besides K, Y, and E are enriched, while H, P, and C are most strongly depleted (Fig. 1b). In SGs, K is not enriched. Y and F are most strongly enriched, while H, L, and P are most strongly depleted (Supplementary Fig. 1b).

To further understand the compositional properties of disordered regions in protein/RNA granules, we calculated dipeptide frequencies. P-bodies and SGs displayed consistent enrichment of GG, PP, SS, and RR dipeptides (Fig. 1c and Supplementary Fig. 1c). Among hetero-dipeptides, RG, RS, and SR are most enriched in P-bodies and SGs, KE, and KR are more strongly enriched in P-bodies than in SGs (Fig. 1c and Supplementary Fig. 1c). Finally, we analyzed the pairwise distances between selected amino acids in the disordered regions of the entire human disordered proteome, in P-bodies and SGs. Same-charged pairs such as K-K and E-E show a marked tendency to cluster locally in sequence in P-bodies and SGs, whereas amino acids with opposite charges do not (Fig. 1d and Supplementary Fig. 1d). This observation is in accordance with interactions between clusters of oppositely charged residues participating in liquid droplet formation[6].

**Lysine-rich condensates are highly dynamic complex coacervates.** The bioinformatic analysis showed that lysine occurs with high frequency in cytosolic protein/RNA granules. As the formation of protein/RNA granules has been connected to LLPS and coacervation of disordered proteins with RNA, we investigated if lysine-rich polypeptide sequences phase separate in the presence of RNA. To this end, we selected the arginine-rich peptides R2 and R3, which contain either two (R2) or three RRASL sequences (R3) and were previously shown to undergo complex coacervation with RNA[12]. In addition, we synthesized

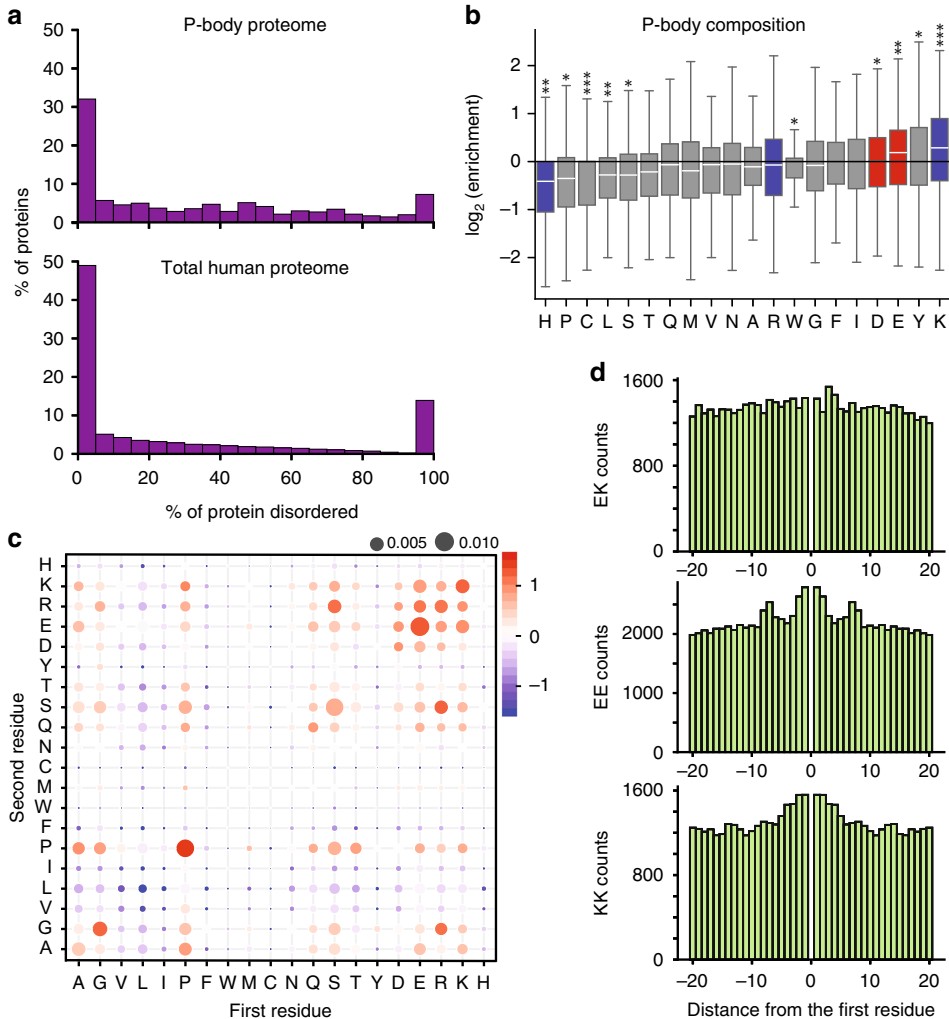

**Fig. 1** Disordered protein regions in P-bodies are rich in lysine. **a** Distribution of the fraction of disorder predicted by IUpred for P-body proteins (top) and the entire human proteome (bottom). **b** Box plot of log2-fold enrichment of amino acids in disordered regions of P-bodies relative to disordered regions of the entire human proteome. Boxes and whiskers show, from bottom to top, 5%, 25%, 50%, 75%, and 95% quantiles. Red/blue: positively/negatively charged residues. $p$-values $< 10^{-5}$:*, $< 10^{-10}$:**, $< 10^{-20}$:***. **c** Log2 enrichment (red) or depletion (blue) of dipeptides in disordered regions of P-bodies relative to disordered regions of the entire human proteome. First residue on $x$-axis, second on $y$-axis. Size of circles: total probability of dipeptide (sum = 1). **d** Same charges cluster together in P-bodies, while opposite charges do not. Top: Total counts of amino acid K at position x relative to an E at position 0 (first residue) in disordered regions of P-bodies. Virtually no clustering is observed. Middle and bottom: Same for K/K and E/E instead of K/E. Residues of same charge are enriched within $+/-$ 5 residues of each other. Source data are provided as a Source Data file

the peptides K2 and K3 that only differ by the type of positively charged side chain, i.e., lysine vs. arginine, and have otherwise identical amino acid sequences when compared to R2 and R3, respectively (Fig. 2a and Supplementary Fig. 2a). We then prepared peptide solutions with transfer RNA (tRNA) or polyU RNA (Fig. 2b) at different peptide concentrations and pH values and found that the mixtures underwent phase separation when K3/R3 concentrations reached ~ 0.5 mM (Fig. 2b, c and Supplementary Fig. 2b). The condensation-transition led to the occurrence of spherical droplets that fused with each other (Fig. 2b and Supplementary Movie 1), indicative for their liquid-like nature[1–3,5]. At equal peptide concentrations R2 phase-separated but not K2 (Supplementary Fig. 2b) and R3 solutions were more turbid and less sensitive to high pH compared to K3 (Fig. 2c). The K3 droplets grew in time at the expense of smaller droplets (Fig. 2d). The number and average size of K3 and R3 droplets increased with increasing tRNA concentration (Fig. 2e and Supplementary Fig. 2c, d), and K3 droplet formation was inhibited upon addition of 30 mM NaCl (Supplementary Fig. 2e), confirming that LLPS

occurs as a complex coacervation between the positively charged K3/R3 and negatively charged RNA chains[1,2,5]. Addition of low concentrations of fluorescently labeled K3 to a solution of pre-formed R3/RNA droplets further demonstrated that lysine-rich sequences can be recruited to arginine/RNA-coacervates (Supplementary Fig. 2f).

Fluorescence recovery after photobleaching (FRAP) showed that the interior of lysine- and arginine-rich droplets is liquid-like (Fig. 2f) and allows rapid exchange of droplet-embedded molecules with the surroundings. In case of R3, the FRAP kinetics were accurately described by a diffusion-controlled recovery model[45] yielding a diffusion coefficient of $D = 0.31$ $\mu m^2 \, s^{-1}$ with a Bayesian information criterion BIC(Axelrod) = $-1075$[46]. As a fit of the experimental data assuming an exponential recovery model resulted in a BIC(exponential) = $-756$, we conclude that the Axelrod model is more appropriate to describe the diffusional properties of R3. In contrast, much faster recovery was observed for K3 and its kinetics were better described by an exponential recovery with a recovery half time of

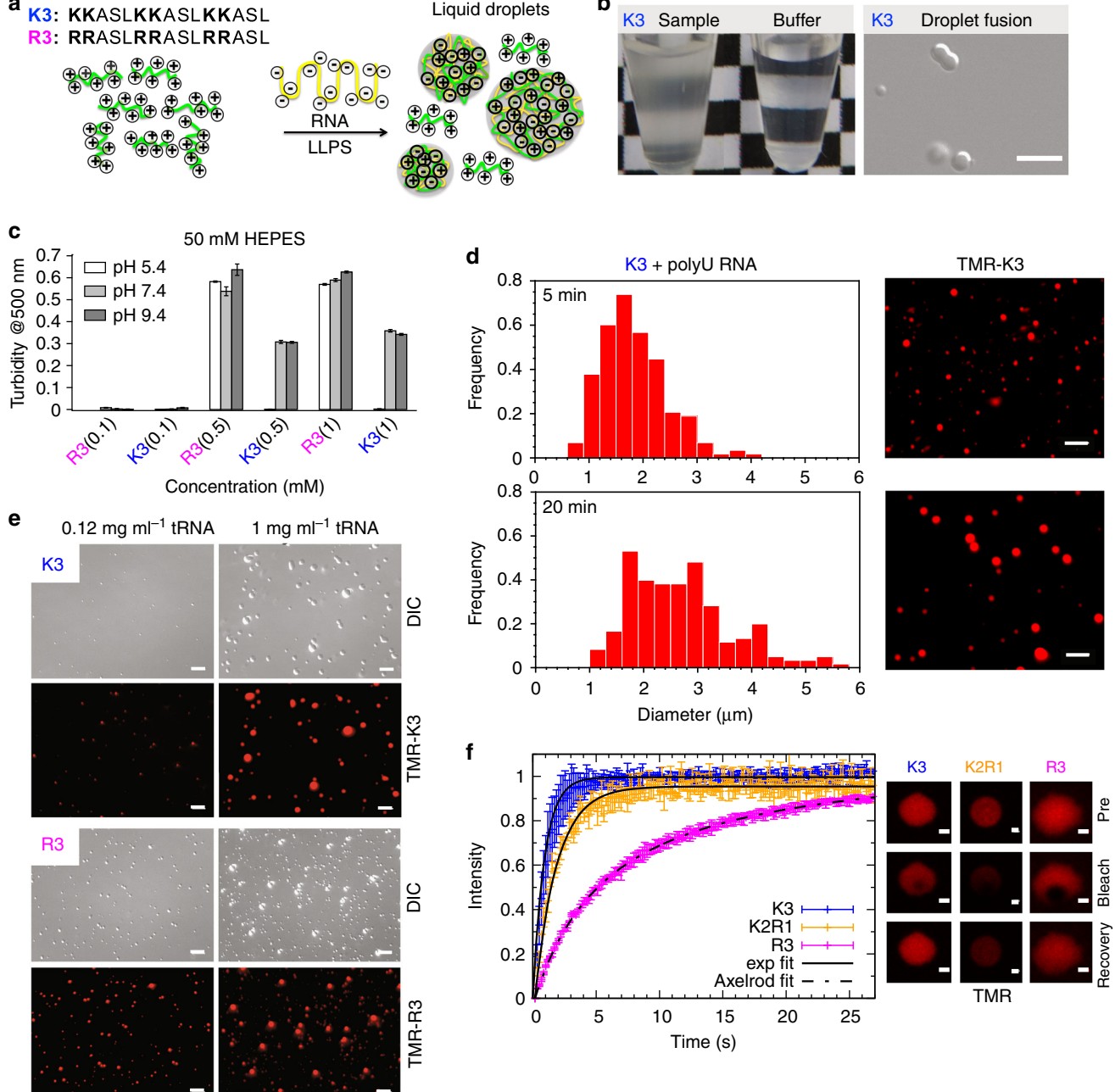

**Fig. 2** Complex coacervation of lysine-rich polypeptides. **a** Amino acid sequences of the lysine-rich (K3) and arginine-rich peptide (R3) along with a schematic of complex coacervation upon addition of RNA. **b** (left) Comparison of the turbid (phase-separated) peptide-containing sample and the control sample/buffer. (Right) DIC image of a pair of K3/RNA liquid droplets undergoing fusion. The concentration of polyU RNA was 0.5 mg ml$^{-1}$ and K3 was 1 mM. **c** Turbidity of K3 with tRNA (0.5 mg ml$^{-1}$) and R3 with tRNA (0.5 mg ml$^{-1}$) samples at different pH values and peptide concentrations (0.1 mM, 0.5 mM, and 1 mM). Reported turbidity values represent the average of a triplicate of measurements for each sample. **d** Distribution of droplet diameters after 5 and 20 min with the corresponding fluorescence images. The concentration of polyU RNA was 1 mg ml$^{-1}$. **e** DIC and fluorescence images depicting the tRNA concentration-dependence of K3 (top) and R3 (bottom) coacervation. Scale bars, 10 μm. **f** Normalized FRAP recovery curves (left) for K3/RNA-coacervates, K2R1/RNA-coacervates and R3/RNA-coacervates with the corresponding FRAP images (right; recovery image 27.2 s after bleaching; scale bar 1 μm). Fits to the experimental data assuming an exponential recovery (K3, K2R1; solid line) and diffusion-controlled recovery (R3; Axelrod model, dash-dotted line). The concentration of K3, K2R1, R3, and polyU RNA in DIC, fluorescence imaging, and FRAP were identical and equal to 1 mM K3/K2R1/R3 and 1 mg ml$^{-1}$ polyU RNA, in 50 mM HEPES, pH 7.4. TMR-K3, TMR-K2R1 and TMR-R3 were used as specified in the methods section. Error bars in **c** and **f** correspond to ± SD. Source data are provided as a Source Data file

0.63 s (Fig. 2f) with a BIC(exponential) = −821, whereas BIC (Axelrod) = −652. To provide further support for the distinct diffusional properties of lysine and arginine-rich polypeptide sequences in RNA-mediated droplets, we prepared the hybrid peptide K2R1, which consists of two KKASL and one RRASL

repeat, and formed droplets through addition of RNA. FRAP analysis showed that the kinetics of fluorescence recovery are in-between those of K3 and R3 (Fig. 2f). In addition, the recovery kinetics of K2R1 were better described by an exponential recovery with a half time of 1.2 s (BIC(exp) = −635), whereas an Axelrod

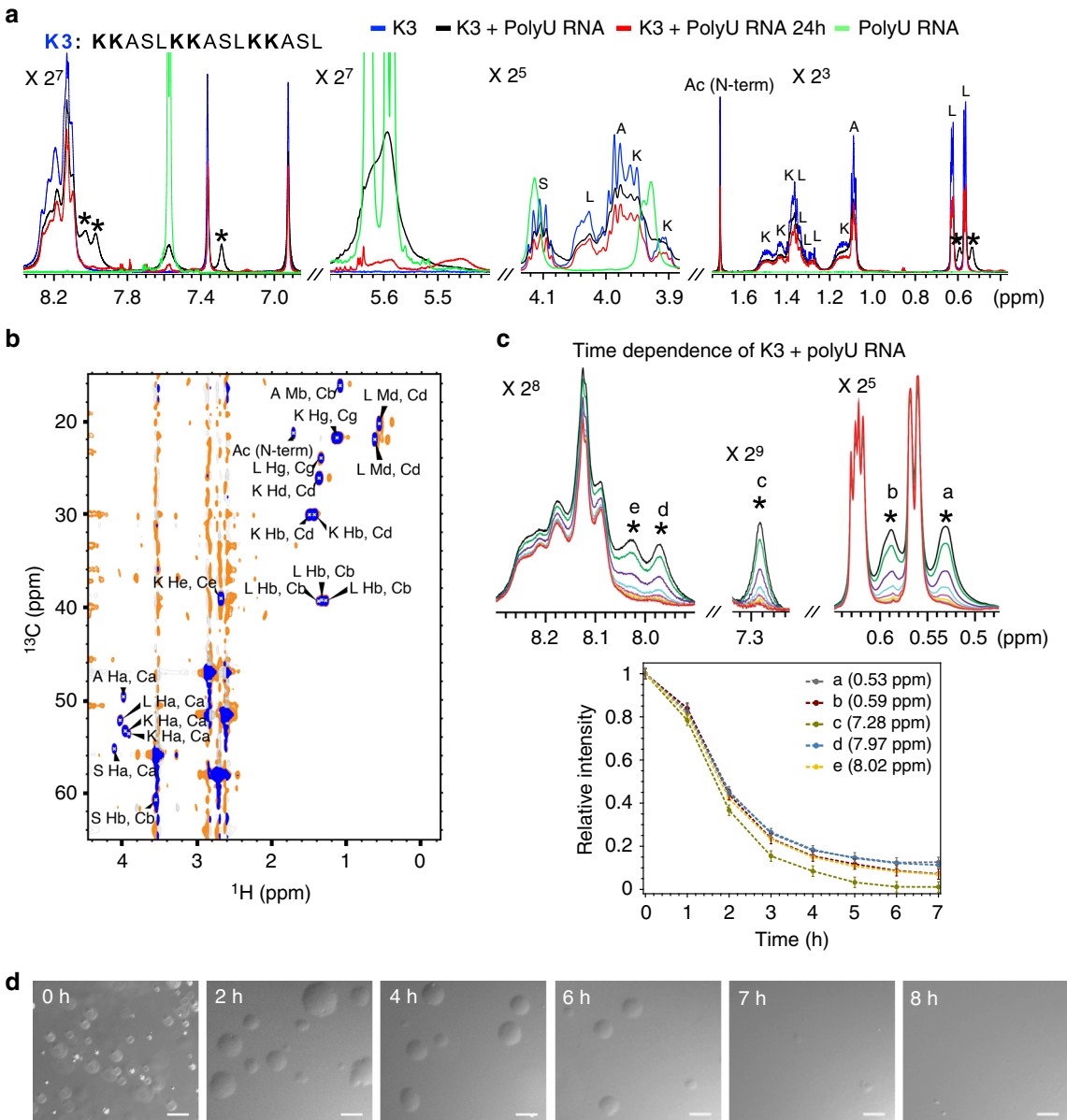

**Fig. 3** NMR spectroscopy and temporal stability of lysine-driven condensation. **a** 1D $^1$H NMR spectra of the lysine-rich peptide K3 (1 mM; blue), K3 with polyU RNA forming liquid droplets (1 mM K3 and 1 mg ml$^{-1}$ polyU RNA; black) and the same sample after 24 h (red), as well as polyU RNA alone (1 mg ml$^{-1}$; green). Peptide NMR signals, which are characteristic for the interaction with polyU RNA, are marked with asterisks. **b** 2D $^1$H−$^{13}$C HSQC NMR spectrum of K3 (blue) and K3 with polyU RNA (orange). Cross-peaks were labeled according to amino acid-specific chemical shift values. The two K3/RNA-specific NMR signals at $^1$H frequencies of 0.53 and 0.59 ppm (marked with asterisks in **a**) were also present in the 2D $^1$H-$^{13}$C HSQC NMR spectrum and have identical $^{13}$C chemical shifts as the two leucine Cδ resonances of K3 in the absence of polyU RNA. **c** Temporal stability of K3 droplets as monitored by the time-dependent decrease of NMR signals. (Top) 1D $^1$H NMR spectra of K3 at 0, 1, 2, 3, 4, 5, 6, and 7 h with K3/RNA-specific signals labeled from **a**–**e**. (Bottom) Time-dependent decrease in normalized signal intensity of K3/RNA-specific signals. Error bars are based on signal-to-noise ratios in the NMR spectra. The peptide solution became clear after ~ 7 h. **d** DIC micrographs of a sample containing K3 (1 mM) and polyU RNA (1 mg ml$^{-1}$) over a period of 8 h. Liquid droplets disappeared after about 7–8 h. Scale bar, 10 µm

fit could not describe the data at times longer than 10 s (BIC (Axelrod) = −598). Thus, the lysine residues appear to dominate the diffusional properties of K2R1. The combined data suggest a more open and less entangled molecular mesh in lysine-rich condensates.

**Lysine/RNA-coacervates differ from arginine/RNA-coacervates**. We then used nuclear magnetic resonance (NMR) spectroscopy to assess the molecular properties of lysine- and arginine-rich coacervates. One-dimensional $^1$H spectra displayed decreased signal intensities and line-broadening upon mixing R3

with RNA (Supplementary Fig. 3a), potentially a result of attenuated motional narrowing upon interaction of the peptide with RNA. LLPS-induced line-broadening in $^1$H-detected NMR spectra was previously observed for several intrinsically disordered proteins[17,34,47,48]. In case of R3/RNA-coacervates, no additional specific features beside signal broadening were detected (Supplementary Fig. 3a). In contrast, new NMR signals were observed when mixing K3 with RNA (Fig. 3a and Supplementary Fig. 3b–d). Resonances specific for lysine/RNA-interactions were detected in the leucine Cδ region near ~ 0.6 ppm, and also at ~7.28 ppm and ~ 8 ppm (see peaks marked with * in Fig. 3a).

Two-dimensional [1]H-[13]C heteronuclear single quantum coherence spectra showed that the two lysine/RNA-characteristic [1]H resonances at ~ 0.6 ppm couple to leucine Cδ resonances (Fig. 3b), indicating that these new resonances are a result of the different chemical environment of the two protons in the free and RNA-bound form. Notably, hydrophobic interactions of leucine residues[49] and hydrogen bonding between serine side-chains and the peptide backbone[50] stabilize helical conformations in short peptides and proteins. In addition, we observed that the specific [1]H signals at ~ 8 ppm coupled to Hα protons in a two-dimensional total correlation spectrum (Supplementary Fig. 3d). Further NMR measurements at different peptide and RNA concentrations, as well as increasing ionic strengths, showed that the new NMR signals are visible in both the dispersed phase (e.g., at low peptide concentration) and the phase-separated state (Supplementary Fig. 4). The observation of peptide signals specific for the interaction with RNA in case of K3 but not R3 suggests that arginine more strongly interacts with RNA when compared to lysine, in agreement with the observed differences of K3 and R3 to form liquid-like droplets in the presence of RNA (Fig. 2).

The intensity of the lysine/RNA-specific NMR signals decayed in time and disappeared after ~ 7 h (Fig. 3c). In parallel, differential interference contrast (DIC) microscopy showed that the number of observable droplets gradually decreased during this time (Fig. 3d). In case these observations would be the result of the reversible dissociation of droplets, it would be expected that the NMR spectra would become similar to those observed at the same peptide/RNA-ratio but an overall lower peptide concentration, i.e., in conditions in which no droplets are formed (Supplementary Fig. 4). Instead we found that both the K3 NMR signals and the polyU RNA resonances decreased. This points to an irreversible process resulting in the sedimentation of droplets or precipitation of K3/RNA-complexes.

**RNA-mediated phase separation of tau sequences**. To gain further insight into the importance of lysine for biomolecular condensation, we investigated the phase behavior of solutions of lysine-rich sequences of the Alzheimer-related protein tau. To this end, we prepared the full-length 441-residue isoform of tau (hTau40), the projection domain of tau comprising the N-terminal 184 residues (termed K25), as well as tau's repeat domain (termed K18; Fig. 4a). Different concentrations of polyU RNA were selected based on the pI of the corresponding tau sequences. We found that all three tau sequences undergo LLPS in a concentration-dependent manner via complex coacervation with polyU RNA (Fig. 4b). hTau40, which contains a total of 44 lysine and 14 arginine residues and has the lowest pI = 8.24 among the three, required the smallest amount of RNA to phase separate (Fig. 4b, left column). The highest concentration of RNA was required for K18, which contains 20 lysines and 1 arginine and has a pI = 9.73 (Fig. 4b, left column; Supplementary Fig. 5a). hTau40 not only phase-separated with RNA, but also without RNA in the presence of dextran as a crowding agent (Fig. 4b, middle column; and Supplementary Fig. 5b) or in HEPES buffer (Fig. 4b, right column), likely because of the ampholytic nature of the amino acid sequence of full-length tau.

Phase-separated droplets of tau formed with polyU RNA were spherical, fused and grew at the expense of smaller droplets (Fig. 4b, c and Supplementary Movie 2 and 3). Phase separation of K18 with RNA was reversed upon increasing the ionic strength of the solution (Supplementary Fig. 5c), while addition of aliphatic alcohols (1,6- or 2,5-hexanediol) did not dissolve tau and K3/RNA droplets (Supplementary Fig. 5e). However, dextran-promoted hTau40 droplets did not grow during the observation phase (Fig. 4c, bottom panel), and salt

concentrations < 100 mM NaCl promoted hTau40 droplets (Supplementary Fig. 5d), suggesting a different mechanism of phase separation when compared to tau/RNA complex coacervation.

To assess the molecular mobility of hTau40 and K18 within droplets, we formed droplets of both proteins using identical concentrations of polyU RNA (Fig. 5a). Subsequent FRAP analysis revealed a diffusion-controlled recovery with a significantly higher diffusion coefficient of K18 ($D$(K18) = 0.007 μm[2] s[−1]) as compared to hTau40 ($D$(hTau40) = 0.002 μm[2] s[−1]) (Fig. 5b). Statistical analysis resulted in BIC(Axelrod) values of −459 and −346 for hTau40 and K18, respectively, whereas BIC(exponential) values were −445 and −289 for hTau40 and K18, respectively. The faster diffusion of K18 within the droplets is consistent with the higher relative abundance of lysine when compared to arginine (15% K and 0.8% R in K18 compared to 10% K and 3% R in hTau40).

**Acetylation reverses lysine-driven liquid–liquid phase separation**. In vivo, lysine is extensively post translationally modified by acetylation[21]. To determine if acetylation influences lysine-driven condensation, we acetylated the lysine-rich peptides and tau sequences with lysine acetyltransferase (KAT) p300 or CREB in the presence of acetyl-Coenzyme A (AcCoA) (Fig. 6a, b and Supplementary Fig. 6a, b). Mass spectrometry identified two and three acetylated groups in K3 (Supplementary Fig. 6a, b), and a mixture of 5-, 6-, 7-, 8-, and 9-fold acetylated K18, and 12 out of 44 acetylated lysines in hTau40 (Fig. 6b).

DIC and fluorescence microscopy revealed that acetylation results in gradual shrinking and ultimately dissolution of coacervates of hTau40 formed in presence of dextran, or of K18 coacervates formed with RNA (Fig. 6c). In case of hTau40, droplets had disappeared already after ~ 10 min, while droplets were no longer visible after ~ 25 min for K18 (Fig. 6c). Quantification of solution turbidity at 500 nm further showed that after 25 min the K18 droplets had not yet fully dissolved, but were too small to be detected by microscopy. After ~ 1 h, K18/RNA solutions were fully transparent (Fig. 6d, right panel). Control experiments in the absence of CREB or in the absence of AcCoA demonstrated that K18/RNA droplets or dextran-promoted hTau40 droplets dissolution was specific for lysine acetylation (Fig. 6d and Supplementary Fig. 6c, d). The specificity of acetylation-dependent complex coacervation was further supported by KAT reactions with lysine- and arginine-rich peptide/RNA-coacervates (Fig. 6e), demonstrating that R3/RNA-coacervates are not dissolved in the presence of CREB or p300.

In order to gain insight into how acetylation interferes with protein/RNA LLPS, we performed NMR titrations of wild-type and acetylated hTau40 (Fig. 7a, b) in the presence of different concentrations of tRNA. We observed pronounced changes in the chemical shifts of many residues in 2D [1]H-[15]N correlation spectra of hTau40 upon increasing concentration of tRNA (Fig. 7c and Supplementary Fig. 7). Sequence-specific analysis revealed that the chemical shift perturbations predominantly occur in the central part of hTau40 from residues ~ 123–386, which contains most of the lysine and arginine residues and carries a net positive charge at pH 7.4 (Fig. 7d). Comparison of chemical shift changes induced by equal concentrations of tRNA in wild-type and acetylated hTau40 demonstrated that tRNA causes smaller perturbations in the acetylated protein (Fig. 7e), which is indicative for a decreased affinity of tRNA to acetylated hTau40. The observation of an attenuated but not inhibited interaction is in agreement with the finding that not all lysine residues of htau40 where acetylated by CREB (Fig. 7a, b). In addition, arginine residues, which are not affected by acetylation, will further contribute to the interaction of hTau40 with tRNA.

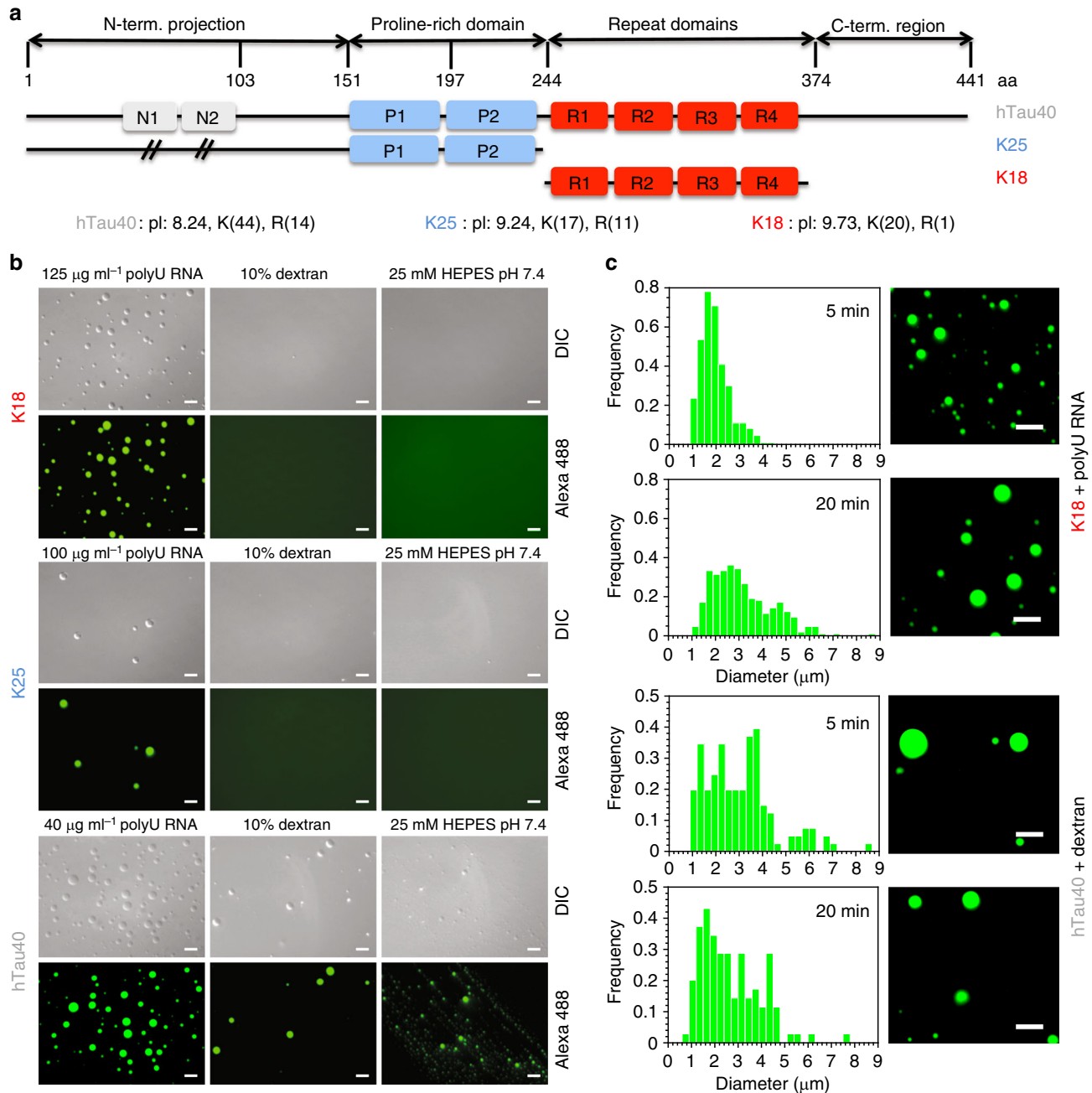

**Fig. 4** Influence of tau domain structure and co-factors on LLPS. **a** Domain organization of hTau40, K25, and K18. N1 and N2 are the two inserts not present in K25, P1, and P2 the two proline-rich regions, and R1–R4 the four pseudo-repeats. The pI and the number of lysine (K) and arginine (R) residues in hTau40, K25 and K18 are indicated. **b** DIC and fluorescence micrographs of hTau40, K25, and K18 solutions in the presence of different co-factors/buffer conditions. The concentration of the proteins was 50 µM. **c** Droplet size distribution of K18 (top) and hTau40 (bottom) after five and 20 min alongside corresponding fluorescence images. K18 (50 µM) with 125 µg ml⁻¹ polyU RNA in 25 mM HEPES, pH 7.4, was used. For hTau40 (50 µM), the same buffer was used with 10% dextran (and no RNA). Scale bar, 10 µm

**Acetylation reduces co-localization of tau with stress granules.**
Tau protein co-localizes with stress granules (SGs) in vivo[38,41,51,52]. To investigate the importance of lysine for co-localization of tau with SGs, we analyzed different tau sequences and acetylated tau for their binding to SGs in semi-permeabilized Hela cells[18] (see schematic in Fig. 8a). Here, SGs were induced by the proteasome inhibitor MG132 and the plasma membrane of cells was subsequently selectively permeabilized by digitonin. Soluble cytosolic factors were washed out and nuclear pore complexes blocked with wheat germ agglutinin (WGA). Next, equal amounts of recombinant, fluorescently labeled tau proteins

were added and after extensive washing, SGs were visualized by immunostaining against the SG marker proteins G3BP1 or TIA-1. Fluorescence microscopy showed that MG132 induced robust formation of SGs, which stained for both G3BP1 and TIA-1 (Fig. 8b–d). In addition, all three tau constructs (hTau40, K25 and K18) co-localized with SGs, but to a different extent (Fig. 8b). Quantification further showed that K25 most strongly associates with SGs (Fig. 8c and Supplementary Fig. 8a). K18 and hTau40 had similar levels of SG co-localization, although slightly higher fluorescence intensity was on average observed for hTau40 when compared to K18 (Fig. 8c). Stronger SG co-localization of hTau40

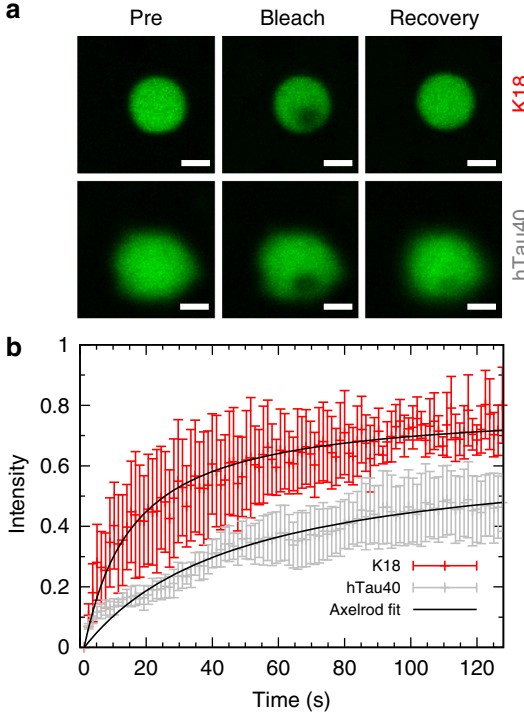

**Fig. 5** Dynamics of tau in tau/RNA droplets. **a** FRAP images (recovery image 129 s after bleaching; scale bar 1 μm). **b** Normalized FRAP recovery curves for hTau40 (50 μM; gray) and K18 (50 μM; red) droplets in the presence of 40 μg ml$^{-1}$ polyU RNA (25 mM HEPES, pH 7.4). Symbols with error bars depict experimental results; black lines are fits assuming diffusion-controlled recovery (i.e., the Axelrod model). Error bars in **b** correspond to ± SD. Source data are provided as a Source Data file

dynamic nature, intrinsically disordered proteins are powerful drivers of LLPS and biomolecular condensation[3,5–8]. The multivalent and cooperative process of LLPS is intimately connected to the amino acid composition of disordered regions in proteins[3,5–8,10]. In addition, the molecular properties of biomolecular condensates and their ability to enable and modulate biochemical reactions are linked to their physico-chemical characteristics and thus to the type and distribution of amino acids in condensed disordered proteins[3,5–7,10].

To gain insight into the physico-chemical properties of biomolecular condensates, we analyzed the amino acid content of cytosolic protein/RNA granules (Fig. 1). Bioinformatic analysis shows that protein disorder is more abundant in proteins associated with SGs and P-bodies than in the entire human proteome (Fig. 1). In addition, the amino acid content in disordered regions of proteins associated with SGs and P-bodies differs from each other, as well as from the human disordered proteome. Lysine is abundant in protein/RNA granules and is the most enriched amino acid in P-bodies when compared to the human disordered proteome (Fig. 1). Lysine can therefore play an important role in the process of biomolecular condensation and modulate the properties of protein/RNA granules.

Supporting a functional role of lysine in biomolecular condensation (Fig. 1), we show that designed lysine-rich peptides phase separate in the presence of RNA (Fig. 2). The resulting liquid-like droplets are more dynamic (Fig. 2) and differ in their molecular properties from arginine/RNA droplets (Fig. 3 and Supplementary Fig. 3). The distinct molecular properties of coacervates formed by lysine- and arginine-rich peptides originate from differences in the chemical structure of the two side chains. Although lysine has a lower pKa (10.5 vs. 13.8 for arginine[54]), both will be fully dissociated at physiological pH and thus carry the same charge. When embedded in a peptide or protein the effective pKa of the arginine residue, however, can potentially be higher due to the larger intrinsic pKa and result in altered dissociation equilibria. In addition, the electron cloud is delocalized across the planar guanidinium group of arginine[55], which introduces directional preferences for cation-pi interactions with aromatic groups, such as tyrosine and phenylalanine, a potential reason for the importance of arginine-tyrosine interactions for LLPS of RNA-binding proteins with prion-like low-complexity domains[7,17,56–58]. The tetrahedral ammonium cation in lysine has a more spherically symmetric charge distribution with weaker directional preferences[55,59]. In addition, the number and nature of hydrogen bonds formed by the ammonium and guanidinium cations with the phosphate anion differ. The side chain of lysine can form more hydrogen bonds than the one of arginine, but in contrast to the guanidinium case with almost perfectly co-linear N–H···O atoms, N–H···O angles with lysine side chains are distorted to 120°[60,61]. Alternatively, lysine forms only one hydrogen bond but with the energetically optimally aligned N–H···O[60]. These effects are expected to lead to stronger interactions of arginine with RNA and can account for the more cohesive nature of arginine-rich condensates as compared to lysine-rich peptides, in agreement with the more rapid diffusion observed in K3 and K2R1 droplets when compared to R3 droplets (Fig. 2f) and a higher mobility of a peptide comprising 30 proline-lysine dipeptides in RNA-induced droplets when compared to a 30 proline-arginine repeat peptide[11]. High-permeability condensates were observed in LAF-1 droplets and were suggested to allow for size-selective filtering analogous to FG nucleoporins in the nuclear pore[62].

The interactions between arginine- and lysine-containing polypeptides and nucleic acids also depend on the length of the peptide, salt concentration, the peptide/nucleic acid molar ratio and the structure of RNA[11,63–66]. Depending on the particular

when compared to K18 was supported through fluorescent labeling of the two native cysteine residues (Cys291 and Cys322) in tau (Supplementary Fig. 8b). The finding that co-localization of tau with SGs decreases in the order K25 > hTau40 > K18 suggests that the repeat region of tau has an inhibitory effect on the co-localization of tau with SGs.

To systematically assess the role of lysine and arginine for co-localization with SGs, we also performed experiments with the model peptides R3, K2R1, and K3 (Supplementary Fig. 9). Addition of the fluorescently labeled peptides to the semi-permeabilized cells showed that their SG co-localization decreases in the order R3 > K2R1 > K3. SG co-localization thus decreased with decreasing arginine content. Notably, the relative abundance of arginine also decreases in the order K25 > hTau40 > K18 (6% > 3% > 0.8%), suggesting a correlation between relative arginine content and SG co-localization.

In vitro experiments showed that acetylation inhibits tau LLPS[53] (Fig. 6). To determine if acetylation also influences co-localization of tau with SGs, we acetylated full-length tau protein and compared it to unmodified hTau40 in the SG association assay (Fig. 8d). In multiple, independent experiments the fluorescence intensity of acetylated hTau40 was comparable to background fluorescence intensity (Fig. 8e), indicating that acetylation strongly reduces association of tau with SGs.

## Discussion

Cells form and use non-membrane-bound compartments/biomolecular condensates to execute and regulate biochemical reactions[1,5]. The formation of protein/RNA granules and other biomolecular condensates depends on weak multivalent interactions between a large number of molecules[5,6]. Because of their

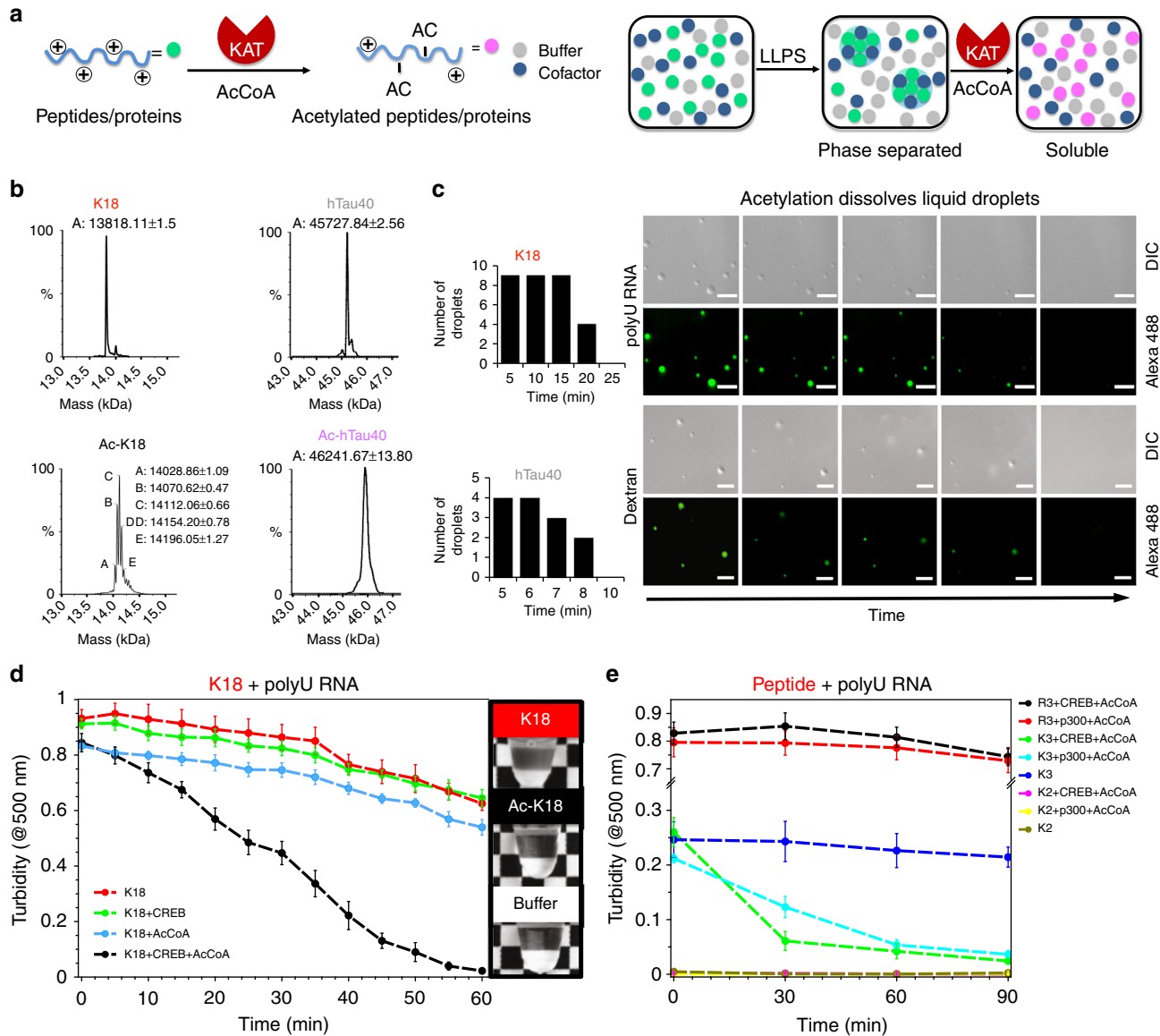

**Fig. 6** Acetylation reverses lysine-driven coacervation. **a** Cartoon representation illustrating acetylation of lysine-rich polypeptides by lysine acetyltransferase (KAT) in the presence of acetyl-Coenzyme A (AcCoA) (left) and its influence on lysine/RNA coacervation (right). **b** Mass spectra of K18 and hTau40 samples before (top) and after acetylation by CREB (bottom). Identified masses (in Da) are indicated. **c** DIC and fluorescence images show dissolution of K18/RNA (top; 50 μM K18, 125 μg ml$^{-1}$ polyU RNA in 25 mM HEPES, pH 7.4) and hTau40/dextran (bottom; 50 μM hTau40, 10% dextran in 25 mM HEPES, pH 7.4) droplets in the presence of the acetyltransferase CREB and AcCoA. Time-dependent changes in droplet numbers were quantified (left). Experiments were performed three times for each protein sample. **d, e** Turbidity measurements of K18 (50 μM; **d**), as well as arginine- (R3; 1 mM) and lysine-rich (K2 and K3; 1 mM) peptides (**e**), which underwent LLPS through addition of polyU RNA (125 μg ml$^{-1}$ for K18, 0.5 mg ml$^{-1}$ for the peptides), in the presence of either CREB/p300 or AcCoA, or both the enzyme and AcCoA. In the middle, photographs of the respective K18 samples after 60 min are depicted. Reported values represent an average of three independent samples. Error bars in **d** and **e** correspond to ± SD. Source data are provided as a Source Data file

experimental conditions these interactions can give rise to reversible non-specific ionic interactions or specific, stoichiometric (i.e., completely charge-neutralizing), cooperative and virtually irreversible interactions that eventually can lead to precipitation[63,64,66], with arginine generally displaying a much stronger precipitation tendency[63,67]. It is therefore conceivable that a competition between 'classical" polyelectrolyte coacervation and condensation driven by specific peptide/RNA-interactions is responsible for the phase behavior observed in the present study, with peptide/RNA-interactions initially driving condensation via complex coacervation, which due to specific cooperative peptide/RNA-interactions eventually lead to gradual precipitation

of an irreversible stoichiometric complex that is invisible in NMR experiments and is observed in terms of a decay of peak intensities characteristic for specific peptide/RNA-interactions (Fig. 3).

Fluorescence microscopy and NMR spectroscopy showed that lysine/RNA-interactions differ from arginine/RNA-interactions resulting in distinct molecular properties in protein/RNA droplets (Figs. 2 and 3). Lysine residues in disordered protein regions can therefore provide a distinct environment in the interior of biomolecular condensates. In cases where arginine/RNA-interactions dominate the process of condensation, lysine would influence the nature of biochemical reactions within the condensate. In addition, combinations of arginine and lysine residues in disordered

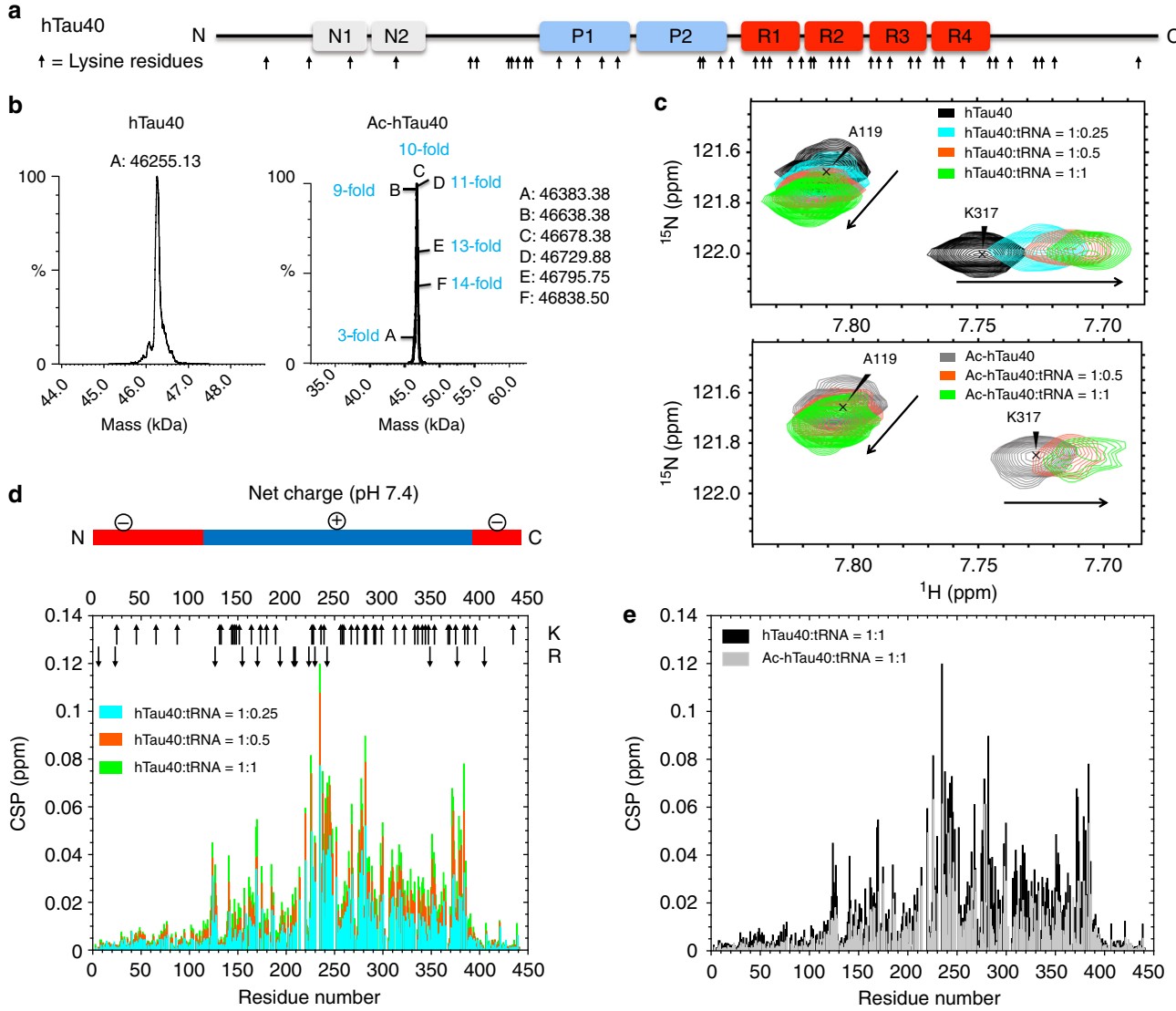

**Fig. 7** Acetylation weakens binding of tau to RNA. **a** Domain organization of hTau40. The location of its 44 lysine residues is marked by arrows. **b** Mass spectra of [15]N-labeled hTau40 (left) and [15]N-labeled hTau40 acetylated by CREB (right). Identified masses (in Da) are indicated. Tau is acetylated at up to 14 sites. **c** Superposition of selected regions of 2D [1]H-[15]N spectra of hTau40 (top, black) and acetylated hTau40 (bottom, gray), respectively, for increasing molar ratios of tRNA: 0.25 (cyan), 0.5 (orange) and 1 (green). **d** Averaged [1]H,[15]N chemical shift perturbations (CSP) of resonances in 2D [1]H-[15]N HSQC spectra of hTau40 caused by binding to tRNA (color code as in **c**). The net charge of different domains of hTau40 is illustrated on top. Arrows mark lysine and arginine residues. **e** CSP of resonances in 2D [1]H-[15]N HSQC of hTau40 (black) and acetylated hTau40 (gray) caused by the addition of an equimolar concentration of tRNA (see also **b**)

regions of condensed proteins can provide a strategy to optimize the affinity for a given protein or nucleic acid, which in pure arginine condensates would have too high affinity.

Previous studies have established a link between the microtubule-associated protein tau and SGs[38,41,51,52]. In addition, LLPS of tau results in supersaturation of tau in liquid-like droplets, which provide an environment that can recruit negatively charged co-factors and promote misfolding and pathogenic aggregation of tau[34,36]. The role of individual domains of tau for cellular condensation has however been largely unknown. We show that wild-type tau can phase separate at high protein concentrations into liquid-like droplets at physiological pH, either alone or in presence of RNA (Fig. 4b). The N-terminal projection domain and the repeat region of tau can also independently phase separate with RNA (Fig. 4b). All three tau sequences contain a large number of lysine residues. In particular, the repeat region

construct K18 has 20 lysine residues but only 1 arginine and can undergo self-coacervation at pH 8.8[34], as well as complex coacervation with RNA at physiological pH (Fig. 4b). All three tau variants associate with SGs when added to semi-permeabilized cells (Fig. 8b). The strongest SG association was observed for K25, which lacks the repeats, followed by hTau40 and K18 (Fig. 8c). This suggest that the repeat region has an inhibitory effect on the SG association of tau. In addition, changes in the binding properties to cytosolic proteins such as microtubules and chaperones could contribute to differences observed in the association of different tau domains with SGs.

Acetylation of lysine side chains is an important cellular modification of proteins[21,23]. We showed that in a synthetic lysine-rich peptide with designed amino acid content, acetylation reverses coacervation with RNA and can thus be specifically associated with modification of the lysine side chain (Fig. 6e). In

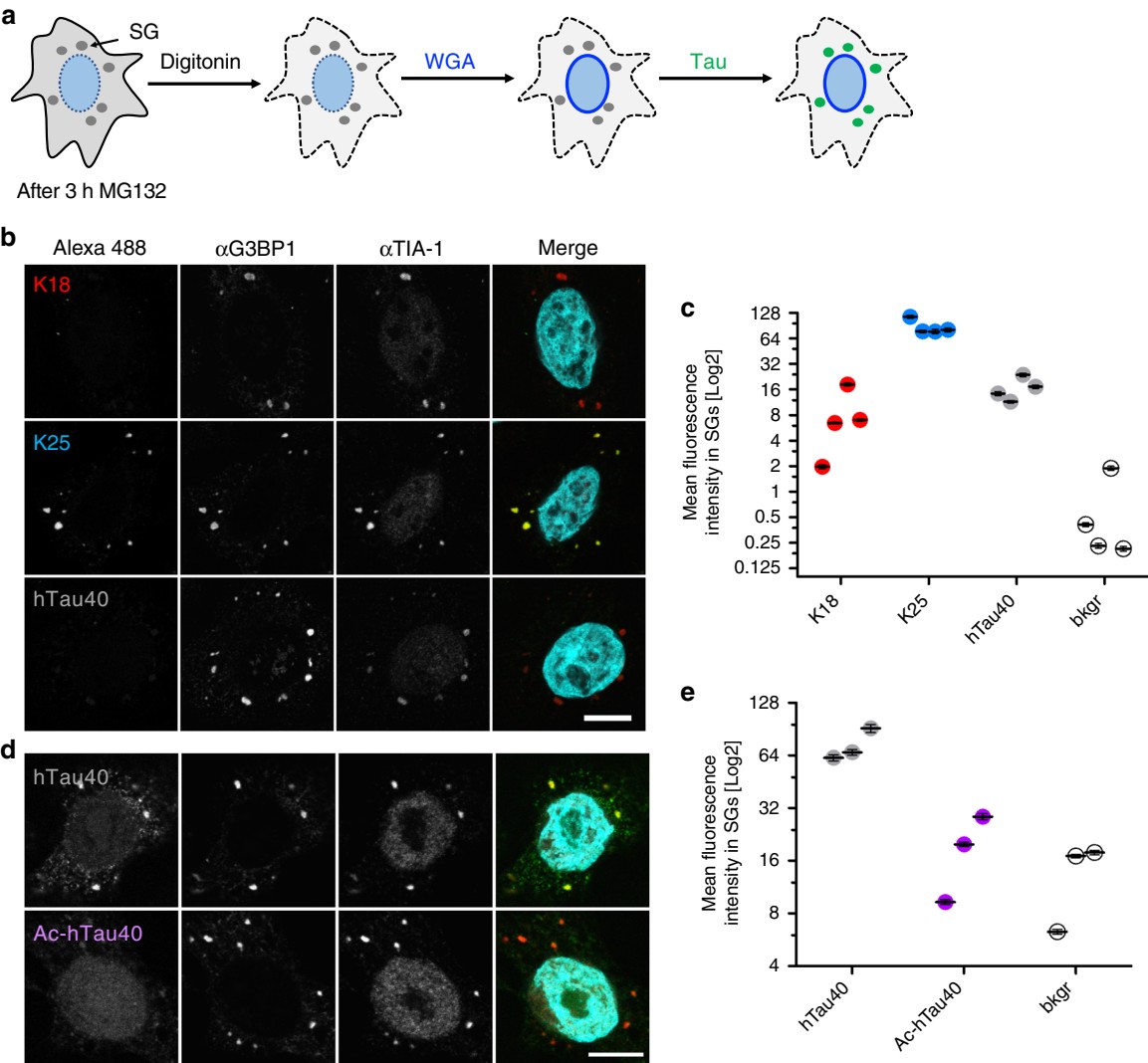

**Fig. 8** Influence of tau domains and lysine acetylation on SG association of tau. **a** Schematic diagram illustrating the semi-permeabilized cell assay. After SG induction by MG132, the plasma membrane of Hela P4 cells is selectively permeabilized using digitonin, cytosolic soluble factors are washed out and subsequently nuclear pore complexes blocked by WGA. Next, Alexa 488-labeled tau proteins are incubated with semi-permeabilized cells to allow binding to SGs and after extensive washing, SGs were subjected to immunostaining for the SG makers G3BP1 and TIA-1. **b** SG association of K18, K25, and hTau40 (400 nM; degree of labeling ("DOL corrected") = 3.7), scale bar, 10 μm. In the merge, Alexa 488 fluorescence (green) and TIA-1 staining (red) are shown. **c** Quantification showing the Log2-transformed mean fluorescence intensity of K18, K25, and hTau40 vs. background (bkgr) in SGs. Four replicates with at least 8–10 cells and ≥ 25 SGs each are shown. Error bars correspond to ± SEM. **d** SG association of unmodified hTau40 (400 nM; DOL = 1.0) in comparison to acetylated hTau40 (Ac-hTau40) monitored by Alexa 488-fluorescence. In the merge, Alexa 488 fluorescence (green) and TIA-1 staining (red) are shown. **e** Quantification showing the Log2-transformed mean fluorescence intensity of hTau40 and Ac-hTau40 vs. background in SGs. Four replicates with at least 8–10 cells and ≥ 37 SGs each are shown. Error bars correspond to ± SEM. Please note, that due to the wide range of fluorescence intensities of the recombinant proteins in SGs, different laser settings had to be used in **b** when compared to **d** in order to avoid pixel saturation and to ensure ideal representation of the differential SG recruitment of the different proteins. Source data are provided as a Source Data file

addition, we demonstrated that acetylation of different regions of tau dissolves liquid-like tau droplets formed in presence of a molecular crowding agent (Fig. 6c), in agreement with previous results[53], but also tau K18 droplets formed through coulombic interactions with RNA (Fig. 6c, d). Besides the frequency and distribution of lysine residues in disordered proteins (Fig. 1), lysine acetylation might thus influence the composition and molecular properties of biomolecular condensates. Consistent with this hypothesis, the lysine acetyltransferase CBP and lysine deacetylases HDAC6 and SIRT6 co-localize with SG proteins in mammalian cells[68–70], pharmacological inhibition or genetic ablation of HDAC6 can abolish SG formation[69], deacetylation of the RNA helicase DDX3X by HDAC6 is necessary for SG

maturation[71] and acetylation of DDX3X interferes with its ability to undergo phase separation[71]. Acetylation of TDP-43, a RNA-binding protein that is found in SGs[72], undergoes LLPS in vitro[8,9,47] and aggregates into insoluble deposits in the brain of patients with amyotrophic lateral sclerosis and frontotemporal lobar degeneration[73], promotes TDP-43 aggregation into SGs, while HDAC6-dependent deacetylation of TDP-43 reduces SG formation[70]. In case of the Alzheimer's disease-related protein tau, we find that its acetylation inhibits LLPS and reduces/suppresses co-localization with SGs (Fig. 8d, e). As acetylation of tau increases tau-induced neurotoxicity[30,33,74], decreased association of acetylated tau with SGs suggests a potential neuroprotective role of SGs in the context of tau-related neurotoxicity. Such a

neuroprotective mechanism could involve the transient trapping of non-acetylated, aggregation-competent tau in SGs.

The ability of many enzymes to post translationally modify lysine residues[23] provides further opportunities to influence and regulate lysine-driven biomolecular condensation. Ubiquitination of lysine is a major determinant for protein degradation[75] and protein deubiquitination by the deubiquitinase enzyme Ubp3 promotes assembly of SGs[76]. Targets of deubiquitinases or ubiquitin-conjugating enzymes might be low-complexity regions of proteins located in SGs and P-bodies, which our bioinformatic analysis have been shown to be rich in lysine residues (Fig. 1). Attachment of ubiquitin chains to lysine residues in disordered regions are expected to sterically interfere with the weak multi-valent interactions that are required for LLPS and biomolecular condensation.

In summary, our integrative approach comprising bioinformatic analysis, in vitro phase separation studies, fluorescence microscopy and NMR spectroscopy, together with a cellular SG association assay, establishes lysine as a critical regulator of biomolecular condensation.

## Methods

**Selection of proteins for bioinformatic analysis**. For analysis of the amino acid content in disordered regions of SGs and P-bodies, protein data sets determined by Jain et al.[42] and Hubstenberger et al.[43] were used, respectively. The total human proteome was collected from the UniProt proteome repository (UP000005640). The human disordered proteome was the collection of sequences predicted to be disordered from the total human proteome. Proteins were predicted to be either ordered or disordered with IUpred[44]. IUpred utilizes pairwise inter-residue interaction energies to define the likelihood of order in a local region. Default parameters were used to obtain both residue-specific probabilities of being ordered and regions of predicted disorder. Both IUpred-predicted ordered and disordered regions were collected and stored for later use with code written in Python version 3.6.3.

**Dipeptide frequency calculation**. Composition is both the simplest and most important feature of protein sequences. To understand the composition of the disordered sequences, the frequency of each of the 400 dipeptides was analyzed using various sets of proteins collected from the Gene Ontology Consortium or other experimental datasets[42,43,77]. Collections were performed with BiomaRt functions written in $R$ version 4.3.3. The resulting FASTA files served as input for the previously mentioned IUpred pipeline, in order to gather sequences predicted to be disordered. Sequences with fewer than 50 residues were discarded and the frequency of each dipeptide in the remaining sequences was measured. To analyze differences between the composition of disordered sequences found within certain sets of proteins, the logarithmic odds ratio (LOR, logarithm base 2) of each dipeptide frequency was calculated. To obtain a dipeptide frequency, the counted observations of each dipeptide was divided by the total number of observations. The $20 \times 20$ matrix was initialized with a pseudo-count of one for each dipeptide.

**Peptide synthesis**. Lysine- (K2: (KKASL)$_2$, K3: (KKASL)$_3$) and arginine-rich peptides (R2: (RRASL)$_2$, R3: (RRASL)$_3$) were synthesized with N-terminal Fmoc protection group chemistry on a Libety1 (CEM) instrument, and purified by HPLC (Reversed-phase, RP18, JASCO). The hybrid peptide K2R1 ((KKASL)$_2$RRASL)) and peptides labeled with tetramethylrhodamine (TMR) at the N-terminus (TMR-K3, TMR-K2R1, and TMR-R3) were synthesized as trifluoroacetic acids salts by GenScript. Peptide stock solutions were made in nuclease-free water (Amresco).

**Protein preparation**. Tau proteins (hTau40, K25, and K18[78]) were expressed in *Escherichia coli* strain BL21(DE3)[78] from a pNG2 vector (a derivative of pET-3a, Merck-Novagen, Darmstadt) in the presence of an antibiotic. In case of unlabeled proteins, the cells were grown in 1–10 l LB and induced with 0.5 mM IPTG at OD$_{600}$ of 0.6–0.8. To obtain $^{15}$N-labeled protein, cells were grown in LB until an OD$_{600}$ of 0.6–0.8 was reached, then centrifuged at low speed, washed with M9 salts (Na$_2$HPO$_4$, KH$_2$PO$_4$, and NaCl) and resuspended in minimal medium M9 supplemented with $^{15}$NH$_4$Cl as the only nitrogen source and induced with 0.5 mM IPTG. After induction, the bacterial cells were harvested by centrifugation and the cell pellets were resuspended in lysis buffer (20 mM MES pH 6.8, 1 mM EGTA, 2 mM DTT) complemented with protease inhibitor mixture, 0.2 mM MgCl$_2$, lysozyme and DNAse I. Subsequently, cells were disrupted with a French pressure cell press (in ice cold conditions to avoid protein degradation). In the next step, NaCl was added to a final concentration of 500 mM and boiled for 20 min making use of the heat stability of the protein. Denatured proteins were removed by ultracentrifugation at 127,000 × g for 40 min at 4 °C. The supernatant was then

put into dialysis tubings (3.5–5 kDa dialysis membrane from Spectra/Por) and dialyzed overnight at 4 °C under constant stirring against dialysis buffer (20 mM MES pH 6.8, 1 mM EDTA, 2 mM DTT, 0.1 mM PMSF, 50 mM NaCl) to remove salt. The following day the sample was filtered and applied onto a previously equilibrated ion exchange chromatography column and the weakly bound proteins were washed out with buffer A (same as dialysis buffer). Tau protein was eluted with a linear gradient of 60% final concentration of buffer B (20 mM MES pH 6.8, 1 M NaCl, 1 mM EDTA, 2 mM DTT, 0.1 mM PMSF). Protein samples were kept and concentrated by ultrafiltration (5 kDa Vivaspin from Sartorius) and purified by gel filtration chromatography. In the last step the protein was dialyzed against 25 mM Hepes pH 7.4, and flash-frozen aliquots were stored. Protein concentrations were determined using a BCA assay.

**Liquid–liquid phase separation**. If not stated otherwise, 1 mM of peptide in 50 mM HEPES, pH 7.4, was used and LLPS was induced by addition of polyuridylic acid potassium salt (polyU RNA, $M_W = 600$–1000 kDa, ~ 2000–3300 residues from Sigma-Aldrich) or transfer RNA from baker's yeast (tRNA, $M_W = 23$–27 kDa, ~76–90 residues from Invitrogen by Thermo Fisher Scientific). In case of LLPS of tau proteins (and if not stated otherwise), 50 µM of protein in 25 mM HEPES, pH 7.4, was mixed with cofactor: 125 µg ml$^{-1}$ polyU RNA in case of K18, 100 µg ml$^{-1}$ polyU RNA in case of K25, 40 µg ml$^{-1}$ polyU RNA or 10% dextran (T500 from Pharmacosmos) in case of hTau40. Experiments with tau proteins were performed at room temperature in reducing conditions in the presence of 1 mM DTT to prevent cysteine oxidation.

**Peptide/protein acetylation**. Peptides were acetylated using CREB (recombinant hCREB binding protein (catalytic domain) from Enzo) and p300 (recombinant hp300 binding protein (catalytic domain) from Enzo). The reaction was performed by mixing 1 mM K3 with 0.62 µM CREB or p300, 1 mM AcCoA, 0.5 mM PMSF, 0.1 mM EDTA in 50 mM HEPES, pH 7.4. For acetylation of K18/hTau40, 50 µM of the protein were mixed with 0.62 µM CREB, 1 mM AcCoA, 0.5 mM PMSF, 0.1 mM EDTA, 1 mM DTT in 25 mM HEPES, pH 7.4. For efficient acetylation, samples were incubated at 30 °C for 2 h. To investigate the effect of acetylation on pre-formed droplets, LLPS was induced by mixing K18 with 125 µg ml$^{-1}$ polyU RNA, hTau40 with 10% dextran or K3 with 1 mg ml$^{-1}$ polyU RNA, followed by enzyme addition.

**Labeling of proteins with fluorescent dyes**. K18, K25, and hTau40 were fluorescently labeled on lysine residues using Alexa Fluor 488 Microscale Protein Labeling Kit (Thermo Fisher Scientific) following vendor's instructions. In case of acetylated hTau40 (Ac-hTau40), the two native cysteine residues (C291, C322) of hTau40 were used for attachment of a fluorescent dye. Before labeling, hTau40 was acetylated and the acetylation enzyme was removed by boiling the sample for 20 min followed by ultracentrifugation. The pellets (denatured enzymes) were removed and the supernatant containing Ac-hTau40 was subjected to fluorescent labeling using Alexa Fluor 488 C5 Maleimide dye (Thermo Fisher Scientific). For labeling, 15 moles of dye were used for each mole of protein. The proteins were incubated in a light-protected Eppendorf tube with the dye freshly prepared in dimethyl sulfoxide (DMSO) for 2 h at room temperature. Excess dye was removed by passing the sample twice through a 0.5 ml 700 MWKO Zeba spin desalting column (Thermo Fisher Scientific). For comparison, hTau40 was also labeled at the two native cysteine residues. Prior to the labeling procedure, DTT was removed from the buffer through dialysis using a slide-A-lyzer mini dialysis device (MWCO 3500, Thermo Fisher Scientific), followed by equilibration in 50 mM HEPES, pH 7.4, with 1 mM TCEP. The degree of labeling (DOL) was determined according to the protocol described in Alexa Fluor 488 Microscale Protein Labeling Kit.

**DIC and fluorescence microscopy**. DIC and fluorescence images were acquired using a Leica DM6000B microscope with a 63x objective (water immersion) and processed using FIJI software (NIH)[79]. For imaging, unlabeled peptide/protein samples were mixed with labeled TMR-labeled peptide (1 µM) or Alexa 488-labeled protein (0.6 µM). At these concentrations, the labeled peptide/protein alone does not form liquid droplets. To estimate the size of liquid droplets, 5–20 DIC images were selected from different sample regions on the slide. Images were then analyzed using FIJI (NIH) by measuring droplet diameters. Size distributions are displayed as normalized frequency histograms.

**Solution turbidity**. Turbidity values of peptide and protein samples were determined at room temperature and a wavelength of 500 nm using a BioSpectrometer Kinetic (Eppendorf) instrument. For samples without enzymes, polyU was the last component added to the sample. In case of K18, the initial turbidity was too high and samples were diluted three times prior to the time-dependent measurement of the turbidity (Fig. 6d).

**Fluorescence recovery after photobleaching**. FRAP experiments were recorded on a Leica TCS SP8 confocal microscope using 63x objective (oil immersion) and a 560-argon laser line. For FRAP measurements, liquid droplets were formed by mixing 1 mM of peptide with 1 µM of TMR-labeled peptide and 1 mg ml$^{-1}$ of

polyU RNA in 50 mM HEPES, pH 7.4, and 50 μM of the protein (hTau40 or K18) with 0.6 μM Alexa 488-labeled protein, 1 mM DTT and 40 μg ml$^{-1}$ polyU RNA. A small region of interest (approx. 1.3 μm for the peptides and 0.6 μm in diameter for the proteins) was chosen and bleached with 30 frames (peptides) or 5 frames (proteins) of full laser power. Fluorescence recovery for the peptides was imaged at low laser intensity of 0.7% and 160 frames were recorded with one frame per 170 ms. In case of the hTau40/K18, the laser intensity was set to 0.4% and 100 frames were recorded with one frame per 1.29 s. Images were analyzed using FIJI (NIH)[79]. FRAP recovery curves were processed, normalized and analyzed in Mathematica by fitting to an exponential recovery as well as the diffusion-controlled recovery model by Axelrod et al.[45] The Bayesian information criterion (BIC)[80] was determined for both models and used to select the most appropriate one. The model with the lowest (i.e., more negative) value was selected. Three FRAP curves were averaged to produce each FRAP curve for analyses.

**NMR spectroscopy.** NMR spectra of the peptides (K3 and R3) and polyU RNA were acquired at 5 °C on Bruker 600 and 800 MHz spectrometers equipped with triple-resonance 5 mm cryogenic probes. Spectra were processed with TopSpin 3.5 (Bruker) and analyzed using Sparky[81]. One-dimensional $^1$H and two-dimensional $^1$H−$^{13}$C heteronuclear single quantum coherence (HSQC) spectra were recorded at natural abundance for uniformly dispersed K3 (1 mM K3 in 50 mM HEPES, 10% D$_2$O), K3 in liquid droplets (1 mM K3, 1 mg ml$^{-1}$ polyU RNA in 50 mM HEPES, 10% D$_2$O) and polyU RNA (1 mg ml$^{-1}$ polyU RNA in 50 mM HEPES, 10% D$_2$O). 2D $^1$H−$^1$H TOtal Correlation SpectroscopY (TOCSY) was recorded to support peptide peak-assignment. 1D $^1$H NMR spectra of R3 (1 mM R3 in 50 mM HEPES, 10% D$_2$O) and R3 in liquid droplets (1 mM R3, 1 mg ml$^{-1}$ polyU RNA in 50 mM HEPES, 10% D$_2$O) were acquired additionally.

NMR titrations of hTau40 with tRNA were acquired at 5 °C on a Bruker 800 MHz spectrometer equipped with a triple-resonance 5 mm cryogenic probe. 2D $^1$H-$^{15}$N HSQC spectra were recorded for the proteins (50 μM) with increasing molar ratios of tRNA (0.25, 0.5 and 1) in 25 mM HEPES, pH 6.8 and 10% D$_2$O. $^{15}$N-labeled hTau40 was acetylated similar to the procedure described above. Here, however, 6 mM AcCoA were used and the sample was incubated at 30 °C overnight. Spectra were processed with TopSpin 3.5 (Bruker) and the peaks were picked and analyzed using Sparky[81]. Chemical shift perturbations (CSP) of individual residues were calculated according to CSP = $[(\delta^1 H_{bound} - \delta^1 H_{free})^2 + 0.1(\delta^{15}N_{bound} - \delta^{15}N_{free})^2]^{1/2}$, considering the relative dispersion of the proton and nitrogen $\delta$ chemical shifts.

**Mass spectrometry.** Mass spectra of acetylated and unmodified peptides and proteins were determined by liquid chromatography (Acquity Arc system, Waters) combined with mass spectrometry (ZQ 4000 Single Quad D2, Alliance).

**Semi-permeabilized cell assay.** HeLa P4 cells[82] were grown on poly-L-lysine coated 12 mm coverslips (No 1.5), permeabilized with 0.003–0.005% digitonin in TPB (20 mM HEPES pH 7.3–7.4, 110 mM KOAc, 2 mM Mg(OAC)$_2$, 1 mM EGTA, 2 mM DTT and 1 μg ml$^{-1}$ each aprotinin, pepstatin and leupeptin). After several stringent washes, nuclear pores were blocked by 15 min incubation with 200 μg ml$^{-1}$ wheat germ agglutinin (WGA) on ice. Cells were then incubated for 30 min at room temperature in the absence (background control) or presence of 400 nM recombinant/purified tau protein (K18, K25, hTau40, or Ac-hTau40) or 5 μM TMR-labeled peptides, respectively, in TPB. Tau proteins to be compared quantitatively had similar DOL (± 3%) or were corrected to achieve similar DOLs by mixing with unlabeled protein of the same concentration ("DOL corrected"). After several stringent washes to remove non-bound protein, cells were fixed and subjected to immunofluorescence for G3BP1 (Proteintech) and TIA-1 (Santa Cruz, C-20; sc-1751) using Alexa 555 (Thermo Fisher Scientific) and Alexa 647-labeled secondary antibodies (Thermo Fisher Scientific), respectively, to visualize SGs. Tia-1 was stained using an Alexa 647-labeled secondary antibody (Thermo Fisher Scientific), while G3BP1 was stained using either Alexa 488 (with TMR-labeled peptides) or Alexa 555 (with tau proteins) secondary antibodies (Thermo Fisher Scientific).

**Immunostaining of SGs.** For semi-permeabilized cell assays, cells were fixed after the last wash in 3.7% formaldehyde/PBS buffer for 7–10 min at room temperature (RT) and permeabilized in 0.5% TX100/ PBS for 5 min at room temperature. Cells were blocked for 10 min in blocking buffer (1% donkey serum in PBS/0.1% Tween-20) and incubated with primary antibodies in blocking buffer for 1–2 h at RT or overnight at 4 °C. Secondary antibodies were diluted in blocking buffer and incubated for 45 min to 1 h at room temperature. Washing steps after antibody incubation were performed with PBS/0.1% Tween-20. DNA was stained with DAPI at 0.5 μg ml$^{-1}$ in PBS and cells mounted in ProLong™ Diamond Antifade.

**Confocal microscopy of SGs.** Confocal microscopy was performed at the Bioimaging core facility of the Biomedical Center with an inverted Leica SP8 microscope, equipped with lasers for 405, 488, 552, and 638 nm excitation. Images were acquired using twofold frame averaging with a 63 × 1.4 oil objective, and an image pixel size of 59 nm. The following fluorescence settings were used for detection:

DAPI: 419–442 nm, Alexa 488: 498–530 nm or 533 nm, Alexa 555: 562–598 nm, Alexa 647: 650–700 nm. Recording was performed sequentially to avoid bleed-through. Alexa 488-labeled tau proteins and TMR-labeled peptides were recorded with hybrid photo detectors (HyD) choosing laser settings for each comparison individually to make best use of the dynamic range of the HyD and avoiding pixel saturation. Dapi, Alexa 555 and Alexa 647 were recorded with a conventional photomultiplier tube.

**Image analysis and processing.** Confocal images were acquired using LAS X (Leica) and processed using Image J software applying linear enhancement for brightness and contrast. At least 8–10 cells and a minimum of 25 SGs were imaged per condition and replicate. For quantitative measurements, equal acquisition settings and processing conditions for respective image sets were applied. For display the raw fluorescence intensity of tau proteins in SGs in the semi-permeabilized cell assay, measured fluorescence values were log transformed to achieve a more balanced spread and displayed using Graph Prism 5.

**Reporting summary.** Further information on research design is available in the Nature Research Reporting Summary linked to this article.

## Data availability
The source data underlying Figs. 1b, 1c, 2c, 2f, 5b, 6d, 6e, 8c, e and Supplementary Figs 1b, 1c, 8a, b, 9 are provided as a Source Data file. Other data are available from the corresponding author upon reasonable request.

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

## Acknowledgements

We thank K. Overkamp (Max Planck Institute for Biophysical Chemistry, Göttingen) for peptide synthesis and mass spectrometry measurements, Prof. Dr. Ralph Kehlenbach to provide us with HeLa P4 cell line, Dr. S. Ambadipudi for discussions, and Dr. M. Rankovic for help with initial microscopy measurements, Dr. D. Kamin for the help with FRAP measurements, Dr. A. Ibanez de Opakua for NMR discussions and Dr. A. Godec for BIC calculations. We thank S. Sohrabi-Jahromi for many discussions, co-supervision of M.G., and help with Fig. 1, Supplementary Fig. 1. We acknowledge the core facility Bioimaging of the BioMedical Center, LMU Munich, for support. M.Z. was supported by the German Science Foundation (Collaborative Research Center 803; projects A11) and by the advanced grant '787679-LLPS-NMR' of the European Research Council. D.D. was funded by the Deutsche Forschungsgemeinschaft (DFG, German Research Foundation; Emmy Noether grant DO 1804/1-1) and under Germany's Excellence Strategy within the framework of the Munich Cluster for Systems Neurology (EXC 2145 SyNergy–ID 390857198). N.R.-G. was supported by the German Science Foundation (Research Grant RE 3655/2-1). M.Z., J.S., D.D. and E.M. were supported by the German Science Foundation through SPP2191 (M.Z. through DFG 71/9-1).

## Author contributions

T.U.-G. performed phase separation experiments, in vitro FRAP measurements and data analysis, and NMR data acquisition and analysis; S.H. performed SG experiments and data analysis; M.P.G. performed bioinformatic analysis; N.R.-G. helped with NMR experiments; M.-S.C.-O. prepared hTau40 and K18; J.B and E.M. prepared K25; J.S. supervised bioinformatic analysis; D.D. supervised SG experiments; T.U.-G. and M.Z. designed the project; T.U.-G., S.H., E.M., J.S., D.D. and M.Z. wrote the paper.

## Additional information

**Competing interests:** The authors declare no competing interests.

