## [Peer Review File · Nature Communications]

Reviewers' comments:

Reviewer #1 (Remarks to the Author):

I've read and re-read this interesting MS a few times. Overall, the work is superbly crafted, systematic and very clear. It has never happened that I do not feel compelled to write a lengthy tome trying to critique the underlying thinking, semantics, or interpretations. I truly have nothing to offer that would have a material impact on improving this MS. It is timely, well thought out, and rigorous. It deserves to be published, perhaps with some minor semantic adjustments. Of interest to the authors is a study posted on bioRxiv that addresses synergistic issues. Please see: <https://www.biorxiv.org/content/early/2018/12/11/492793>

Reviewer #2 (Remarks to the Author):

Ukmar-Godec and co-workers report that the lysine and its acetylation could regulate biomolecular condensation (or liquid-liquid phase separation, LLPS), a mechanism explaining the spatiotemporal control of biochemical reactions in cells. They started from bioinformatics analysis and found that lysine is the dominant amino-acid type in stress granules. Following this finding, they designed synthetic peptides of different lysine or arginine repeats and demonstrated that lysine has different LLPS property comparing to arginine albeit they both are basic amino acids. They also applied NMR spectroscopy to compare the difference between arginine- and lysine-repeated peptides interacting with RNA molecules. They then used protein tau (with three different constructs), having highly dominant lysine residues, as an example, to demonstrate its interaction with RNA and LLPS property. They also used enzymes that acetylate lysine to demonstrate the condensate can be disrupted via acetylation. Finally, they used a cell model to demonstrate that acetylated tau protein delocalizes with stress granule.

Protein tau, a protein well-studied in its aggregation/disease relation, undergoes LLPS. Its LLPS property has only been recently studied (from the Zweckstetter group (as this manuscript), Nature Comm 2017 8:275 and from the Bradley Hyman's group EMBO Journal 2018 e98049). Although Ferron's group has reported that acetylation disrupts tau's phase separation (2018 Int. J. Mol. Sci. E1360), this manuscript further using a cell model to verify the effect of acetylation. In general, this paper is systematically and logistically demonstrated how lysine and RNA regulate LLPS, and I recommend its publication. However, this article's novelty is aiming to demonstrate a general mechanism that lysine/RNA interaction regulates biomolecular condensation. Thus, could the authors provide other examples other than protein tau? Or could the model peptides (studies in Figure 2-5) also be used in their cell model (Figure 6)? Besides, I also have the following questions:

(1) In the "Lysine is enriched in disordered regions of cytosolic protein/RNA granules" section Several analysis details are missing. I can find their citations about SG and P-bodies proteins, but what is "the disordered proteome" they used to compare? How did the percentages (35% and 25%) determine? (For example, a protein is mostly disordered, then it is counted into the denominator as one, or, a protein with residue numbers from 340 to 410 have an IUpred score higher than (a certain value), this ~70 residues are counted in the denominator). This should be clarified in the method section.

(2) Please comment on using the alanine-serine-leucine tripeptide as linkers to link di-arg or di-lys peptides. (why not use simple linker such as GGG?)

(3) In figure 3b, it was observed that some NMR signals disappear after 7 hours, indicating the dissolve of K3/RNA coacervates. Is this scenario also observed under a microscope or just turbidity? i.e. the droplet disappears after 7 hours.

(4) In figure 6b, the colocalization of K18 and hTau40 is not clear. It is hard to see the signal of K18 and hTau40 in the first column of Figure 6b. Furthermore, how do the authors confirm that the expression level for all these three constructs in cells is the same? Is there any internal control? Is it possible that K18 and hTau40 expressed less in the cells and thus less in SG?

(5) In figure 6d, the Ac-hTau40 looks like colocalized with SG by looking at the three spots on the left-upper side of the nucleus and the two spots on the right-hand side of the nucleus, although they are similar to the background (from the description in the main text and figure 6e). Again, does the Ac-hTau40 or hTau40 expressed at a similar level in those cells? Furthermore, in figure 6c, the log₂-transformed mean fluorescence of hTau40 is around 16 but is 64 in figure 6c. The value of Ac-hTau40 is 16, close to the background, but the background is lower than 1 in 6c. This should be clarified.

Minor:

(1) In Figure 1b, 1c, and 1d

The tick marks and labels (on x- and y-axes) are not properly aligned.

(2) The term DIC and NMR should be defined in their first appearance for more general readers.

(3) There is no legend for Figure 3d.

(4) In Figure 4a, K18 has 28 lysine? Please confirm, I only counted 20.

Reviewer #3 (Remarks to the Author):

The molecular determinants of phase separation is the focus of many current studies (see Wang et al. *Cell*, 2018, Vernon et al. *eLife*, 2018). Here, the authors focus on determining whether lysine residues mediate LLPS in tau. The authors first conduct a bioinformatics study and determine that lysine residues are overrepresented in processing-body (P-body) proteins. Therefore, they focus on the role of lysine residues in phase separation of complex coacervates. Using synthetic peptides, they find that lysine-RNA coacervates are more dynamic than arginine-RNA ones. Furthermore, post-translational modification of lysines into acetyllysines inhibits phase separation, as expected if complex coacervation is mediate by electrostatics. They then find that acetylation of tau reduces tau's ability to phase separate in vitro, as well as exogeneous tau's recruitment to stress granules, membraneless bodies thought to form via liquid-liquid phase separation mechanisms. They conclude that lysine is an important regulator of condensates. These findings are not unexpected given recent studies that confer the importance of post-translational modifications on phase separating proteins, particularly PTMs that change charge (e.g. acetylation, citrullination, phosphorylation). Indeed, as the authors note, reduction of tau phase separation by acetylation has already been recognized (Ferreon et al. *Int J Mol Sci.*, 2018).

My enthusiasm for this work was diminished early due to multiple errors in Figure 1 and figure legends for Figure 1, Figure S1, and Figure 3. For example in Figure 1, the x-axis amino acids in panels b and c are offset. This issue also occurs in panel d. The figure legend for Figure S1 reads identical to that of Figure 1, except for the first sentence. I understand that Figure S1 analyzes amino acid composition of stress granules, whereas Figure 1 focuses on p-bodies, but that is not obvious from reading Figure S1's legend. Figure 3 is missing a description for panel d entirely. Aside from those issues, the results are timely, but are not unexpected from a physicochemical point of view as I remarked above. This needs to be acknowledged. Results are not suitable for publication currently. My concerns are as follows:

Bioinformatic Analysis

Are there particular proteins in P-bodies with high lysine enrichment? It would be helpful to supplement Figure 1 with a list of top protein sequences to see if there are particular ones that bias the bioinformatics analysis. From Figure 1a, I don't understand the basis for the statement "Whereas 25% of amino acids in the total human proteome are predicted to be disordered, the percentage is 35% for the SG proteome and 38% for P-bodies".

Synthetic Peptides

What was the basis for the R2, R3, K2, K3 sequences? Why were those chosen? Also, a third or fourth peptide could be helpful, i.e. for Figure 2f. Here, the authors use R3/RNA and K3/RNA coacervates to comment on dynamics. What if they used a R3/K3 hybrid sequence? Would FRAP

curves be intermediate between R3 and K3? The use of this peptide could assist in determining the role of these residue types in regulating dynamics of coacervates. Did the FRAP curves change over time (i.e. droplet maturation)?

NMR

The authors comment extensively on the new NMR signals that arise only in the presence of K3/RNA coacervates that are not present in R3/RNA coacervates (Figure 3a). However, aren't these new peaks a result of RNA:K3 interactions? If I understand correctly, the K3 reference spectra do not include RNA. When RNA is added, coacervates form and these peaks appear, but they disappear over the next 7 hours. Furthermore, the authors find the sample to be clear again after 7 hours. So I have a couple of comments. Can the authors rule out that these "LLPS-specific" peaks are not peptide:RNA interaction peaks? Why didn't the broadened peaks become sharper if the K3/RNA-coacervates had dissolved after 7hr? If the coacervates do indeed disappear after 7 hours, the authors should be able to recapitulate those conditions *in vitro* by microscopy to confirm. Alternatively, the coacervates could interact with the surface of the NMR tube and gel-ify along the wall of the NMR tube. The authors should confirm that this is not the case. Indeed, they could also perform the acetylation experiment in the NMR tube and probe the interactions between the peptide and RNA in those experiments, as acetylation inhibits LLPS. This would significantly improve the manuscript.

Tau

The focus here is on the lysine-rich sequences of tau, particularly K18. While the authors show this part of the protein phase separates with RNA, the authors could also perform FRAP experiments to examine the liquidity among these different constructs (e.g. K25 and K18). This could also back up several claims regarding lysine and arginine's role in regulating dynamics of coacervates (as in the synthetic peptide section).

Acetylation

The authors show that lysine acetylation decreases LLPS of synthetic peptides with RNA, as well as tau with RNA. Are there particular lysines in tau that are acetylated preferentially *in vivo*? They use KAT and CREB here, but are these the most relevant enzymes for tau acetylation? Understanding the mechanism behind how acetylation disrupts tau recruitment to stress granules would strengthen the manuscript. Is it through a reduction of interactions between tau and RNA, or other protein-protein interactions with tau?

Stress granules

Figure 6b. In the microscopy image for hTau40, it does not appear that tau colocalizes with stress granules, whereas it does in Figure 6d. Can this be fixed?

Minor concerns

Color scheme in figure 1. Generally, acidic and basic residues are colored red and blue, respectively. This is reversed in panel b. (Same holds for Figure S1)

Figure 5a: Acetylation of lysines removes the positive charge (if at pH 7), but does not make the lysine negatively charged as depicted in the cartoon.

In the discussion section, arginine's pKa is considerably higher than 12.5 in aqueous solutions. Please see (Fitch et al. Protein Science, 2015).

We thank the reviewers for their comments and have taken this opportunity to strengthen our manuscript. Please find below a detailed response made in view of the reviewers' suggestions.

Reviewer: 1

Comments to the Author

I've read and re-read this interesting MS a few times. Overall, the work is superbly crafted, systematic and very clear. It has never happened that I do not feel compelled to write a lengthy tome trying to critique the underlying thinking, semantics, or interpretations. I truly have nothing to offer that would have a material impact on improving this MS. It is timely, well thought out, and rigorous. It deserves to be published, perhaps with some minor semantic adjustments. Of interest to the authors is a study posted on bioRxiv that addresses synergistic issues. Please see: <https://www.biorxiv.org/content/early/2018/12/11/492793>

Reply: We are grateful to the reviewer for the positive assessment of our work. We would also like to thank the reviewer for pointing out this very interesting study, which we included into the revised version of the manuscript (pages 3 and 11 and new reference 11).

Reviewer: 2

Comments to the Author

Ukmar-Godec and co-workers report that the lysine and its acetylation could regulate biomolecular condensation (or liquid-liquid phase separation, LLPS), a mechanism explaining the spatiotemporal control of biochemical reactions in cells. They started from bioinformatics analysis and found that lysine is the dominant amino-acid type in stress granules. Following this finding, they designed synthetic peptides of different lysine or arginine repeats and demonstrated that lysine has different LLPS property comparing to arginine albeit they both are basic amino acids. They also applied NMR spectroscopy to compare the difference between arginine- and lysine-repeated peptides interacting with RNA molecules. They then used protein tau (with three different constructs), having highly dominant lysine residues, as an example, to demonstrate its interaction with RNA and LLPS property. They also used enzymes that acetylate lysine to demonstrate the condensate can be disrupted via acetylation. Finally, they used a cell model to demonstrate that acetylated tau protein delocalizes with stress granule. Protein tau, a protein well-studied in its aggregation/disease relation, undergoes LLPS. Its LLPS property has only been recently studied (from the Zweckstetter group (as this manuscript), Nature Comm 2017 8:275 and from the Bradley Hyman's group EMBO Journal 2018 e98049). Although Ferron's group has reported that acetylation disrupts tau's phase separation (2018 Int. J. Mol. Sci. E1360), this manuscript further using a cell model to verify the effect of acetylation. In general, this paper is systematically and logistically demonstrated how lysine and RNA regulate LLPS, and I recommend its publication.

Reply: We are grateful to the reviewer for the positive assessment of our work and for the suggestions, which we addressed in detail and revised our manuscript accordingly.

However, this article's novelty is aiming to demonstrate a general mechanism that lysine/RNA interaction regulates biomolecular condensation. Thus, could the authors provide other examples other than protein tau? Or could the model peptides (studies in Figure 2-5) also be used in their cell model (Figure 6)?

Reply: To further support the generality of lysine/RNA-driven interactions in biomolecular LLPS we performed, according to the reviewer's suggestion, additional experiments on model peptides, which in the revised version also include an intermediate/hybrid peptide K2R1. In addition to FRAP experiments with K2R1 as well as hTau40 and K18 (please see revised Fig. 2f and newly added discussion on page 6 highlighted in yellow), in which we quantified the relative goodness-of-fit for both theoretical models by means of the Bayesian information criterion, we also performed systematic SG-recruitment experiments of all three model peptides in the cell assay (please see the new Supplementary Fig. 9 and newly added discussion on the bottom of page 9 that is highlighted in yellow).

Briefly summarized, FRAP analysis confirmed the role of lysine residues for the mobility in droplets, by revealing that the results of the recovery for the new hybrid peptide K2R1 lies in-between those of K3 and R3 and were fitted better with an exponential recovery as in the case of K3. Moreover, we also performed FRAP experiments with the proteins hTau40 and K18 and found a diffusion-controlled recovery with a higher diffusion coefficient of K18 as compared to hTau40 (please see new Fig. 5b). The faster diffusion of K18 within the droplets is consistent with the larger relative abundance of lysine when compared to arginine (15 % K and 0.8 % R in K18 compared to 10 % K and 3 % R in hTau40).

Furthermore, the addition of the fluorescently-labeled peptides to the semi-permeabilized cells showed that their SG colocalization decreases in the order R3>K2R1>K3 – i.e. with decreasing arginine content. Notably, the relative abundance of arginine also decreases in the order K25>hTau40>K18 (6 % > 3 % > 0.8 %), which agrees with the decrease of colocalization of proteins with SGs, thus suggesting a correlation between relative arginine content and SG colocalization.

(1) In the “Lysine is enriched in disordered regions of cytosolic protein/RNA granules” section Several analysis details are missing. I can find their citations about SG and P-bodies proteins, but what is “the disordered proteome” they used to compare? How did the percentages (35% and 25%) determine? (For example, a protein is mostly disordered, then it is counted into the denominator as one, or, a protein with residue numbers from 340 to 410 have an IUpred score higher than (a certain value), this ~70 residues are counted in the denominator). This should be clarified in the method section.

Reply: To further clarify how the analysis was performed, we added to the methods section (page 13): *“The total human proteome was collected from the UniProt proteome repository (UP000005640). The human disordered proteome was the collection of sequences predicted to be disordered from the total human proteome.”*

The percentages in fact directly reflect the number of disordered residues (i.e. those predicted to be in a disordered configuration) divided by the total number of residues in a given protein. To clarify this, we included an explanatory sentence on page 4 in the revised version: *“To see the relative importance of disordered regions and their distribution across sequences, the percent of residues predicted to be disordered in each protein in the respective proteomes was plotted.”*

(2) Please comment on using the alanine-serine-leucine tripeptide as linkers to link di-arg or di-lys peptides. (why not use simple linker such as GGG?)

Reply: In our study we selected the arginine-rich peptides R2 and R3, which contain either two (R2) or three RRASL sequences (R3) and were previously shown (Nat. Chem. 8, 129-137 (2016)) to undergo complex coacervation with RNA. Based on this, we synthesized the peptides K2 and

K3 that only differ by the type of positively charged side chain, i.e. lysine vs arginine, and have otherwise identical amino acid sequences when compared to R2 and R3, respectively. To clarify our choice of model peptides, we included the following sentence on page 5 in the revised version of the manuscript: *“To this end, we selected the arginine-rich peptides R2 and R3, which contain either two (R2) or three RRASL sequences (R3) and were previously shown to undergo complex coacervation with RNA [11].”*

(3) In figure 3b, it was observed that some NMR signals disappear after 7 hours, indicating the dissolve of K3/RNA coacervates. Is this scenario also observed under a microscope or just turbidity? i.e. the droplet disappears after 7 hours.

Reply: This observation was indeed confirmed using the microscope (please see new Fig. 3d). In the revised version of the manuscript, we also added an additional paragraph to the discussion section in order to address these observation (page 7).

4) In figure 6b, the colocalization of K18 and hTau40 is not clear. It is hard to see the signal of K18 and hTau40 in the first column of Figure 6b. Furthermore, how do the authors confirm that the expression level for all these three constructs in cells is the same? Is there any internal control? Is it possible that K18 and hTau40 expressed less in the cells and thus less in SG?

Reply: We thank the reviewer for pointing this out. In order to resolve the issue, we have linearly enhanced the brightness and contrast of the black-and-white images for better visibility, using same settings for corresponding channels. Please note that in this assay there is no cellular expression of K18 or hTau40, but that we use recombinant proteins added to semi-permeabilized cells at equal concentrations (400 nM). This is stressed in the figure legend and material & methods section in the revised version of the manuscript. In addition, we rephrased the corresponding text passage in the result section to make this more clear.

(5) In figure 6d, the Ac-hTau40 looks like colocalized with SG by looking at the three spots on the left-upper side of the nucleus and the two spots on the right-hand side of the nucleus, although they are similar to the background (from the description in the main text and figure 6e). Again, does the Ac-hTau40 or hTau40 expressed at a similar level in those cells? Furthermore, in figure 6c, the log₂-transformed mean fluorescence of hTau40 is around 16 but is 64 in figure 6c. The value of Ac-hTau40 is 16, close to the background, but the background is lower than 1 in 6c. This should be clarified.

Reply: As stated above in our reply to comment 4, we add equal amounts of recombinant proteins to individual reactions in this assay and can hence ensure the comparability of ac-hTau40 with hTau40. Please note that each subpanel (i.e. new Fig. 8b and 8d) was acquired using a different laser setting on the confocal microscope due to the limited dynamic range of the confocal detector device and to avoid pixel saturation: The laser intensities were always chosen according to the reaction showing the strongest SG association in individual comparisons (i.e. K25 in panel b, but hTau40 in panel d). Therefore, intensities of hTau40 in subpanel b (and values in subpanel c, respectively) cannot be directly be compared with intensities in subpanel d (or values in e, respectively). Hence, also the background measurements differ between these two data sets. We have now added an explanatory sentence to the legend and material & method section to make this more clear to the reader.

Minor:

- (1) In Figure 1b, 1c, and 1d the tick marks and labels (on x- and y-axes) are not properly aligned.
- (2) The term DIC and NMR should be defined in their first appearance for more general readers.
- (3) There is no legend for Figure 3d.
- (4) In Figure 4a, K18 has 28 lysine? Please confirm, I only counted 20.

Reply: We corrected the minor issues in the revised version of the manuscript.

Reviewer: 3

Comments to the Author

The molecular determinants of phase separation is the focus of many current studies (see Wang et al. Cell, 2018, Vernon et al. eLife, 2018). Here, the authors focus on determining whether lysine residues mediate LLPS in tau. The authors first conduct a bioinformatics study and determine that lysine residues are overrepresented in processing-body (P-body) proteins. Therefore, they focus on the role of lysine residues in phase separation of complex coacervates. Using synthetic peptides, they find that lysine-RNA coacervates are more dynamic than arginine-RNA ones. Furthermore, post-translational modification of lysines into acetyllysines inhibits phase separation, as expected if complex coacervation is mediated by electrostatics. They then find that acetylation of tau reduces tau's ability to phase separate in vitro, as well as exogenous tau's recruitment to stress granules, membraneless bodies thought to form via liquid-liquid phase separation mechanisms. They conclude that lysine is an important regulator of condensates. These findings are not unexpected given recent studies that confer the importance of post-translational modifications on phase separating proteins, particularly PTMs that change charge (e.g. acetylation, citrullination, phosphorylation). Indeed, as the authors note, reduction of tau phase separation by acetylation has already been recognized (Ferreon et al. Int J Mol Sci., 2018). My enthusiasm for this work was diminished early due to multiple errors in Figure 1 and figure legends for Figure 1, Figure S1, and Figure 3. For example in Figure 1, the x-axis amino acids in panels b and c are offset. This issue also occurs in panel d. The figure legend for Figure S1 reads identical to that of Figure 1, except for the first sentence. I understand that Figure S1 analyzes amino acid composition of stress granules, whereas Figure 1 focuses on p-bodies, but that is not obvious from reading Figure S1's legend. Figure 3 is missing a description for panel d entirely.

Reply: We sincerely apologize for the formatting problems of Figs. 1 and S1. The formatting problems have been corrected in the revised version of the manuscript. In addition, the figure legends of Figs. 1, S1 and 3 have been double checked and corrected.

Aside from those issues, the results are timely, but are not unexpected from a physicochemical point of view as I remarked above. This needs to be acknowledged. Results are not suitable for publication currently.

Reply: In the revised version of the manuscript, we have added additional experimental data and further clarifications. In addition, we extended the discussion section in particular addressing the issues raised below. We would also like to point out that maybe some of the results are expected from a physicochemical point of view, some others are clearly not (e.g. high

abundance of lysine in disordered regions of P bodies, decreased colocalization of acetylated tau with stress granules, ...).

Bioinformatic Analysis

Are there particular proteins in P-bodies with high lysine enrichment? It would be helpful to supplement Figure 1 with a list of top protein sequences to see if there are particular ones that bias the bioinformatics analysis. From Figure 1a, I don't understand the basis for the statement "Whereas 25% of amino acids in the total human proteome are predicted to be disordered, the percentage is 35% for the SG proteome and 38% for P-bodies".

Reply: Thanks for the suggestion. We added the full, ordered list of sequences as supporting tables (Supplementary Tables 1-3). Please note that we did not select the sequences entering the analysis according to some preference. In other words, the statistics should reflect a realistic and unbiased result averaged over the respective genome. Even if there were for some reason isolated sequences with an abnormally high lysine enrichment, their statistical weight would be expected to be insignificant.

The percentages do not follow from Fig. 1a, which reflects the percentage of proteins having a given fraction of predicted disorder. They are the direct result of the data analysis and are not depicted again in Fig. 1. In order to clarify this, we included an explanatory sentence in the revised version on page 4: *"To see the relative importance of disordered regions and their distribution across sequences, the percent of residues predicted to be disordered in each protein in the respective proteomes was plotted (Fig.1a, Supplementary Fig.1a) "*.

Synthetic Peptides

What was the basis for the R2, R3, K2, K3 sequences? Why were those chosen? Also, a third or fourth peptide could be helpful, i.e. for Figure 2f. Here, the authors use R3/RNA and K3/RNA coacervates to comment on dynamics. What if they used a R3/K3 hybrid sequence? Would FRAP curves be intermediate between R3 and K3? The use of this peptide could assist in determining the role of these residue types in regulating dynamics of coacervates. Did the FRAP curves change over time (i.e. droplet maturation)?

Reply: The R3 and R2 peptides were previously used to study RNA-induced coacervation of arginine-rich sequences (Nat. Chem. 8, 129-137 (2016)). To allow direct comparison with these previous studies, we chose the same sequences and then replaced the arginine residues by lysine. To clarify our choice of these model peptides, we included the following sentence on page 5 in the revised version of the manuscript: *"To this end, we selected the arginine-rich peptides R2 and R3, which contain either two (R2) or three RRASL sequences (R3) and were previously shown to undergo complex coacervation with RNA [11]."*

As suggested by the reviewer, we added data for a hybrid R3/K3 sequence into the revised version of the manuscript. We selected the sequence K2R1, which consists of two KKASL repeats and one RRASL repeat. We then studied the K2R1 peptide by FRAP as well as in the stress granule colocalization assay. Both the FRAP curve (orange data in Fig. 2f) and the degree of colocalization with stress granules (new Supplementary Fig. 9) of K2R1 were intermediate between R3 and K3, in support of the different roles of arginine and lysine in RNA/protein-coacervation.

To justify the choice of the kinetic model to describe the FRAP dynamics, we further used the Bayesian information criterion as a goodness-of-fit criterium (J. Am. Stat. Assoc. 90, 773-795 (1995)), which confirms the distinct recovery mechanisms of lysine-rich and arginine-rich peptides.

The experiments were performed every 30 min and after 90 min we did not observe a significant change in the FRAP curves over time. The analyzed data represent the average of 3 measurements performed in 30 min intervals. A new paragraph to describe the experiment has been included in Methods section page 16.

NMR

The authors comment extensively on the new NMR signals that arise only in the presence of K3/RNA coacervates that are not present in R3/RNA coacervates (Figure 3a). However, aren't these new peaks a result of RNA:K3 interactions? If I understand correctly, the K3 reference spectra do not include RNA. When RNA is added, coacervates form and these peaks appear, but they disappear over the next 7 hours. Furthermore, the authors find the sample to be clear again after 7 hours. So I have a couple of comments. Can the authors rule out that these "LLPS-specific" peaks are not peptide:RNA interaction peaks? Why didn't the broadened peaks become sharper if the K3/RNA-coacervates had dissolved after 7hr? If the coacervates do indeed disappear after 7 hours, the authors should be able to recapitulate those conditions in vitro by microscopy to confirm. Alternatively, the coacervates could interact with the surface of the NMR tube and gel-ify along the wall of the NMR tube. The authors should confirm that this is not the case. Indeed, they could also perform the acetylation experiment in the NMR tube and probe the interactions between the peptide and RNA in those experiments, as acetylation inhibits LLPS. This would significantly improve the manuscript.

Reply: We thank the reviewer for these comments/suggestions. To address them, we performed a number of additional experiments. First, we monitored the sample in a time-dependent manner by microscopy, demonstrating disappearance of droplets after ~7 h (new Fig. 3d). To minimize the influence of gel-ifying along the wall of the NMR tube, we siliconized the NMR tube prior to addition of the K3/RNA mixture and droplet formation. We then followed the NMR signals of this sample in a time-dependent manner and observed an identical time-dependent attenuation of NMR signals:

Figure. The influence of siliconization of the NMR tube on K3/RNA characteristic NMR peaks. NMR tubes were modified using sigmacote, a commercially-available siliconizing reagent (Merck, Germany), which forms a thin hydrophobic film on glass and prevents adsorption of positively-charged biomolecules. After drying under N₂ gas, the NMR tubes were filled with the sigmacote solution and incubated at room temperature for one hour. The solution was then removed from the NMR tubes, washed with acetone, dried again under N₂ gas and the tubes were kept closed until further use. 1D ¹H NMR spectra of the K3 droplets (1 mM K3 + 1 mg ml⁻¹ polyU RNA in 50 mM HEPES, pH 7.4; black). NMR signals characteristic to the K3/RNA interactions are marked with asterisks. The time-dependent experiment revealed the same behavior of the phase separated sample as in the non-treated NMR tube suggesting that the peptide is not interacting with the NMR tube or gel-ify along the wall of the NMR tube.

Next, we performed additional NMR measurements at different peptide and RNA concentrations, as well as increasing ionic strengths (new Supplementary Fig. 4). The new measurements showed that the new NMR signals are visible in both the dispersed phase (e.g. at low peptide concentration) and the phase-separated state (new Supplementary Fig. 4). The observation of peptide signals specific for the interaction with RNA in case of K3 but not R3 furthermore suggested that arginine more strongly interacts with RNA when compared to lysine, in agreement with the observed differences of K3 and R3 to form liquid-like droplets in the presence of RNA (Fig. 2). Finally, we also recorded NMR spectra of the K3 peptide in presence of AcCoA, CREB and polyU RNA, in order to investigate the influence of acetylation on the NMR spectrum of K3 in presence of RNA. In agreement, with the discussion above the peptide/RNA interaction peaks were not observed (Supplementary Fig. 4d). The new data show that the new NMR signals observed for the K3 peptide in presence of RNA are indeed peptide/RNA interaction peaks as suggested by the reviewer. At the same time, they are attenuated in a time-dependent manner, which correlates with the disappearance of droplets as evidenced by microscopy (Fig. 3d). This points to an irreversible process resulting in the sedimentation of droplets or precipitation of K3/RNA-complexes. We implemented the new observations by (i) adjusting in the Results section the text describing the peptide NMR data, and (ii) including an additional paragraph in the Discussion section, which puts our observations in context with previous reports about gradual precipitation of irreversible peptide/RNA complexes.

Tau

The focus here is on the lysine-rich sequences of tau, particularly K18. While the authors show this part of the protein phase separates with RNA, the authors could also perform FRAP experiments to examine the liquidity among these different constructs (e.g. K25 and K18). This could also back up several claims regarding lysine and arginine's role in regulating dynamics of coacervates (as in the synthetic peptide section).

Reply: As suggested by the reviewer, we have performed FRAP experiments for two different tau constructs (Fig. 5). The results are described on page 8 of the revised manuscript:

"To assess the molecular mobility of hTau40 and K18 within droplets, we formed droplets of both proteins using identical concentrations of polyU RNA (Fig. 5a). Subsequent FRAP analysis revealed a diffusion-controlled recovery with a significantly higher diffusion coefficient of K18 ($D(K18)=0.007 \mu\text{m}^2 \text{s}^{-1}$) as compared to hTau40 ($D(h\text{Tau}40)=0.002 \mu\text{m}^2 \text{s}^{-1}$) (Fig. 5b). Statistical analysis resulted in BIC(Axelrod) values of -459 and -346 for hTau40 and K18, respectively, whereas BIC(exponential) values were -445 and -289 for hTau40 and K18, respectively. The faster diffusion of K18 within the droplets is consistent with the higher relative abundance of lysine when compared to arginine (15 % K and 0.8 % R in K18 compared to 10 % K and 3 % R in hTau40)."

Acetylation

The authors show that lysine acetylation decreases LLPS of synthetic peptides with RNA, as well as tau with RNA. Are there particular lysines in tau that are acetylated preferentially *in vivo*? They use KAT and CREB here, but are these the most relevant enzymes for tau acetylation? Understanding the mechanism behind how acetylation disrupts tau recruitment to stress granules would strengthen the manuscript. Is it through a reduction of interactions between tau and RNA, or other protein-protein interactions with tau?

Reply: The importance of acetylation for tau neurotoxicity has been investigated in several studies, however, little is still known about the relative importance of different acetylation sites (and to which degree the different sites are acetylated) *in vivo*. On the other hand, *in vitro* acetylation with the enzymes CREB and p300 (which we used in our studies) is well

characterized in the literature. Indeed, CREP and p300 were already used in several studies involving acetylation of tau (see e.g. Cohen et al, *Nat. Commun.* (2011) 2, 252 and Min et al, *Neuron* (2010) 67, 953–966). Both enzymes acetylate several lysine residues of tau.

The specific interactions involved in the binding of tau proteins to SGs, and in particular their disruption by means of acetylation, unfortunately could not be addressed using the semi-permeabilized cell assay: the stress granules did not persist upon addition of RNAase, i.e. the stress granule marker proteins G3BP1 and TIA-1 were lost upon incubation of the semi-permeabilized cells with increasing concentrations of RNase:

Figure. The influence of RNase on SG marker proteins. The SG markers G3BP1 and TIA-1 are lost from SGs in semi-permeabilized cells upon RNase treatment. HeLa P4 cells were grown on poly-L-lysine coated 12 mm coverslips (No 1.5), permeabilized with 0.003-0.005% digitonin in TPB (20 mM HEPES pH 7.3-7.4, 110 mM KOAc, 2 mM Mg(OAc)₂, 1 mM EGTA, 2 mM DTT and 1 µg ml⁻¹ each aprotinin, pepstatin and leupeptin). After extensive washing, nuclear pores were blocked by 15 min incubation with 200 µg/ml wheat germ agglutinin (WGA) on ice. Cells were then incubated for 30 minutes at room with increasing concentrations of RNase A (0/ 10/200 µg ml⁻¹) in TPB. After several stringent washes, cells were fixed and subjected to immunofluorescence for G3BP1 (Proteintech) and TIA-1 (Santa Cruz, C-20; sc-1751) using Alexa 555 and Alexa 488 secondary antibodies (Thermo Fisher Scientific), respectively, to visualize SGs. DNA was counterstained using DAPI and cells mounted in prolong diamond antifade (Thermo Fisher Scientific). Confocal imaging was performed as described in the manuscript using PMTs. The scale bar corresponds to 20 µm.

To gain insight into how acetylation interferes with protein/RNA LLPS, we performed NMR-titration experiments of unmodified and acetylated hTau40 (new Fig. 7a, b) in the presence of different concentrations of tRNA. We observed pronounced changes in the chemical shifts of many residues in 2D ¹H-¹⁵N correlation spectra of hTau40 upon increasing concentration of tRNA (please see new Fig. 7c and Supplementary Fig. 7). Sequence-specific analysis revealed that the chemical shift perturbations predominantly occur in the central part of hTau40 from residues ~123-386, which contains most of the lysine and arginine residues and carries a net positive charge at pH 7.4 (Fig. 7d). Comparison of chemical shift changes induced by equal concentrations of tRNA in unmodified and acetylated hTau40 demonstrated that tRNA causes smaller perturbations in the acetylated protein (Fig. 7e), which is indicative for a decreased affinity of tRNA to acetylated hTau40. The observation of an attenuated but not inhibited interaction is in agreement with the finding that not all lysine residues of htau40 where

acetylated by CREB (please see Fig. 7a, b). In addition, arginine residues, which are not affected by acetylation, will contribute to the interaction of hTau40 with tRNA. The new data are presented in a new figure (Fig. 7). In addition, a new paragraph has been included in the revised version of the manuscript on page 9 and the experiments are described in the Methods section on page 17.

Stress granules

Figure 6b. In the microscopy image for hTau40, it does not appear that tau colocalizes with stress granules, whereas it does in Figure 6d. Can this be fixed?

Reply: Please note that each subpanel (i.e. new Fig. 8b and 8d) was acquired using a different laser setting on the confocal microscope due to the limited dynamic range of the confocal detector device and to avoid pixel saturation: In Fig. 8b (previously 6b), the laser settings on the confocal microscope were chosen according to the protein showing the strongest SG association, i.e. K25. Due to the limited dynamic range of confocal detectors, this results in hT40 intensity being close to background level. When we adjust the laser settings to visualize hT40 intensities in SGs, we adjust the dynamic range of the detector to weaker signal intensities and can therefore display hT40 vs ac-hT40 and background. To clarify this further we have revised the Methods section (Confocal microscopy of SGs) on page 18 by including the following text: *“Recording was performed sequentially to avoid bleed-through. Alexa 488-labeled tau proteins and TMR-labelled peptides were recorded with hybrid photo detectors (HyD) choosing laser settings for each comparison individually to make best use of the dynamic range of the HyD and avoiding pixel saturation. Dapi, Alexa 555 and Alexa 647 were recorded with a conventional photomultiplier tube.”*

Minor concerns

Color scheme in figure 1. Generally, acidic and basic residues are colored red and blue, respectively. This is reversed in panel b. (Same holds for Figure S1)

Figure 5a: Acetylation of lysines removes the positive charge (if at pH 7), but does not make the lysine negatively charged as depicted in the cartoon.

In the discussion section, arginine’s pKa is considerably higher than 12.5 in aqueous solutions. Please see (Fitch et al. Protein Science, 2015).

Reply: We corrected the minor issues in the revised version of the manuscript.

REVIEWERS' COMMENTS:

Reviewer #2 (Remarks to the Author):

I am Reviewer #2. The authors have addressed all the comments and suggestions. I think the revised version is very much improved and I recommend its publishing.

Reviewer #3 (Remarks to the Author):

The authors have done an excellent job in addressing my concerns through additional experiments, revised figures, and clearer text throughout methods, results, and discussion. I recommend publication of this work in Nature Communications. Their observations of lysine enrichment in proteins associated with P-bodies, coupled with the observations that lysine/RNA coacervates adopt more liquid-like characteristics than arginine/RNA coacervates are important. The study suggests implications for how liquidity of RNA/protein condensates could be regulated in cells through recruitment of lys-rich or arg-rich proteins. The additional K2R1 peptide experiments provide further insight. The NMR and microscopy data that reveal "disappearing" droplets are quite interesting, and further experiments are necessary to understand this phenomenon (but are outside the scope of this paper). Lastly, the experiments with tau back up observations made with synthetic peptides. The results are timely. I have a few minor recommendations:

- 1) Abstract suggestion: Lysine is enriched in disordered regions OF PROTEINS in P-bodies
- 2) Fig 3D – It would be helpful to image the same area of the coverslip over time.
- 3) Is there a reason why liquid phase separation is used rather than liquid-liquid phase separation?
- 4) Pg. 3 "Lysine with its lysyl ((CH₂)₄NH₂) side chain is a basic and positively charged amino acid at physiological pH."
- 5) A full NMR spectrum of full-length tau would be useful in supplementary info.

Reviewer #3 (Remarks to the Author):

The authors have done an excellent job in addressing my concerns through additional experiments, revised figures, and clearer text throughout methods, results, and discussion. I recommend publication of this work in Nature Communications. Their observations of lysine enrichment in proteins associated with P-bodies, coupled with the observations that lysine/RNA coacervates adopt more liquid-like characteristics than arginine/RNA coacervates are important. The study suggests implications for how liquidity of RNA/protein condensates could be regulated in cells through recruitment of lys-rich or arg-rich proteins. The additional K2R1 peptide experiments provide further insight. The NMR and microscopy data that reveal "disappearing" droplets are quite interesting, and further experiments are necessary to understand this phenomenon (but are outside the scope of this paper). Lastly, the experiments with tau back up observations made with synthetic peptides. The results are timely. I have a few minor recommendations:

- 1) Abstract suggestion: Lysine is enriched in disordered regions OF PROTEINS in P-bodies
- 2) Fig 3D – It would be helpful to image the same area of the coverslip over time.
- 3) Is there a reason why liquid phase separation is used rather than liquid-liquid phase separation?
- 4) Pg. 3 “Lysine with its lysyl ((CH₂)₄NH₂) side chain is a basic and positively charged amino acid at physiological pH.”
- 5) A full NMR spectrum of full-length tau would be useful in supplementary info.

Reply: We thank reviewer #3 for the further suggestions. We have implemented suggestions 1 (now corrected), 3 (now replaced all by liquid-liquid), 4 (corrected) and 5 (added in the SI) in the revised version of the manuscript.

Regarding point Fig. 3D: Because droplets disappear not in a fully synchronized manner, i.e. in one region of the cover slip droplets might disappear faster than in another region, we believe that it is important to select different regions (as done in Fig. 3D), in order to better illustrate the overall decrease in droplet numbers with increasing incubation time.